# THE CURIOUS CASE OF ADAMW

## ABSTRACT

AdamW is ubiquitous in deep learning, yet its behavior remains poorly understood. We analyze its dynamics through the lens of dynamical systems and show that AdamW admits an *implicit objective*: its fixed points coincide with the stationary points of a constrained and regularized optimization problem. However, not all of these fixed points are stable under AdamW's dynamics, and stability depends sensitively on curvature, weight decay, and momentum parameters. Even in simple one-dimensional settings, AdamW can exhibit surprisingly complex behavior: equilibria may be unstable and trajectories can fall into persistent limit cycles. We further extend the analysis to higher dimensions, deriving sufficient conditions for stability, and validate empirically that when AdamW converges in neural network training, it converges to stable equilibria. These results clarify what optimization problem AdamW is associated with, when convergence can be expected, and how its curious dynamics could inspire the development of more reliable optimization algorithms in the future.

## 1 INTRODUCTION

The Adam optimizer with decoupled weight decay (AdamW; (Loshchilov & Hutter, 2017)) has been the workhorse of modern deep learning. From large language models (Grattafiori et al., 2024; Liu et al., 2024), to vision architectures (Ravi et al., 2024), to generative models (Esser et al.), AdamW powers today's most influential systems and is widely regarded as the default optimizer for deep learning.

Yet beneath this ubiquity lies a fundamental gap: while the convergence of Adam (without weight decay; (Kingma & Ba, 2014)) is already subtle to analyze, *no general convergence proof is available for AdamW*[1]. The difficulty arises from the use of decoupled weight decay, which is essential in practice for stability and generalization (Kosson et al., 2023; D'Angelo et al., 2024), but remains disconnected from the objective, misaligned with the gradient, and fundamentally alters the algorithm's dynamics.

**Our contributions.** We present two central findings that clarify AdamW's behavior: what it implicitly optimizes, and how the stability of its dynamics informs its convergence.

- **Implicit objective of AdamW.** We show that the fixed points of AdamW coincide exactly with the stationary points of a constrained and regularized optimization problem.
- **Stability of AdamW.** We further show that not all stationary points of this implicit objective correspond to stable equilibria of AdamW. This gap enables simple counterexamples where AdamW is non-convergent, failing even on benign static objectives.

**Implicit objective.** Contrary to the common view that weight decay acts as "adaptive regularization," we argue that an optimizer's objective should depend only on parameters, not on internal states. By analyzing AdamW's fixed-point conditions, we identify an implicit objective $F$ whose stationary points coincide AdamW's equilibria. This construction connects to a special case of the Lion-$\mathcal{K}$ algorithm (Chen et al., 2023a) and extends the conclusions of Xie & Li (2024). However, the equivalence is only first-order: AdamW does not necessarily minimize $F$, since some local minima of $F$ may be unstable under AdamW's dynamics, and the algorithm may fail to converge to them.

---

[1]See section 6 for discussion.

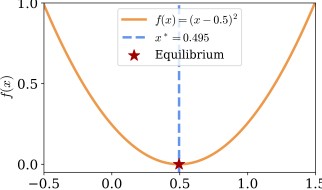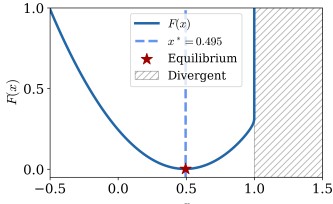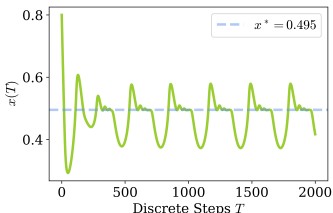

Figure 1: AdamW with $\eta = 10^{-2}$, $\beta_1 = 0.99$, $\beta_2 = 0.9$, $\lambda = 1$, $e = 10^{-2}$. Left: quadratic objective $f(x) = (x - \frac{1}{2})^2$. Middle: implicit objective $F(x)$ whose stationary point coincides with AdamW's equilibrium. Right: AdamW iterates $x(T)$ over time, showing non-convergence despite the simple convex objective.

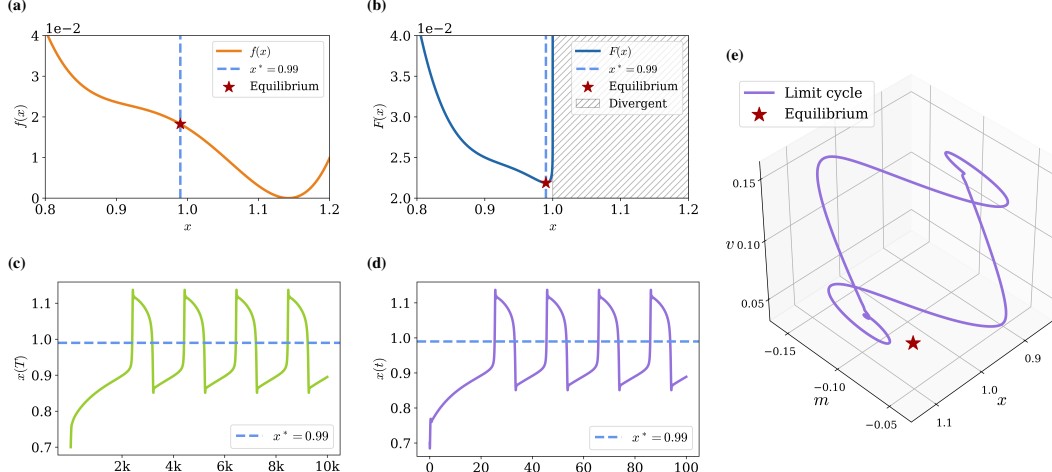

Figure 2: AdamW with $\eta = 10^{-2}$, $\beta_1 = 0.9$, $\beta_2 = 0.99$, $\lambda = 1$, $e = 10^{-2}$. (a) Target objective $f(x)$ with fixed point at $x^* = 0.99$. (b) Implicit objective $F(x)$ whose stationary point coincides with the AdamW fixed point. (c) Discrete-time trajectory $x(T)$. (d) Continuous-time counterpart $x(t)$. (e) Phase portrait in $(x, m, v)$ space. Together, (c–e) show that AdamW does not converge but instead enters a stable limit cycle around the equilibrium.

**Stability of AdamW.** Our analysis shows that stability and stationarity need not coincide for AdamW. This misalignment creates a stability gap that produces explicit counterexamples to convergence: when $\beta_1 > \beta_2$, AdamW diverges even on the simple quadratic $(x - \frac{1}{2})^2$, regardless of step size (see figure 1); in the common regime $\beta_1 < \beta_2$, convergence also fails, as iterates on simple one-dimensional polynomials settle into stable limit cycles, orbiting indefinitely rather than converging (see figure 2). These behaviors are not artifacts of adversarial hyperparameters or pathological online objectives (cf. Reddi et al., 2019), but intrinsic to AdamW's update rule.

We study these dynamics through the lens of stability analysis. If AdamW converges, it necessarily converges to a local minima of the implicit objective, but sufficiency requires additional constraints on the Hessian of the original objective. If the only equilibrium in the domain is unstable, convergence is impossible and trajectories spiral into oscillations. In one dimension, when $\beta_1 > \beta_2$, stability requires an upper bound on curvature, which is often violated even for quadratics; when $\beta_1 < \beta_2$, stability requires a lower bound, so convergence occurs only if the Hessian is not too negative. In higher dimensions the situation is more involved, but local Lyapunov analysis yields tractable sufficient conditions. For neural networks, the Hessian is too large to characterize exactly, yet experiments reveal a consistent pattern: whenever AdamW converges, it does so at a stable equilibrium. Whether this requirement for stability explains AdamW's empirical success or conceals hidden pitfalls remains open, but what is clear is that convergence is not guaranteed, and the curious dynamics of AdamW open intriguing directions for future research.

---

**Algorithm 1** AdamW Update Rules [2]

---

**Require:** initial parameters $x_0$, hyperparameters $\{\eta_t\}_{t=1}^T, \beta_1, \beta_2, e$, weight decay strength $\lambda > 0$
 1: Initialize moment $m_0 \leftarrow 0, \quad v_0 \leftarrow 0$
 2: **for** $t = 1, 2, \ldots, T$ **do**
 3:     Compute gradient $\boldsymbol{g}_t \leftarrow \nabla f_t(\boldsymbol{x}_{t-1})$
 4:     Update moment: $\boldsymbol{m}_t \leftarrow \beta_1 \boldsymbol{m}_{t-1} + (1-\beta_1)\boldsymbol{g}_t, \quad \boldsymbol{v}_t \leftarrow \beta_2 \boldsymbol{v}_{t-1} + (1-\beta_2)\boldsymbol{g}_t^{\odot 2}$
 5:     Bias correction: $\hat{\boldsymbol{m}}_t \leftarrow \boldsymbol{m}_t(1-\beta_1^t), \quad \hat{\boldsymbol{v}}_t \leftarrow \boldsymbol{v}_t(1-\beta_2^t)$
 6:     Parameter update:

$$\boldsymbol{x}_t \leftarrow \boldsymbol{x}_{t-1} - \eta_t \frac{\hat{\boldsymbol{m}}_t}{\sqrt{\hat{\boldsymbol{v}}_t} + e} - \eta_t \cdot \lambda \boldsymbol{x}_{t-1}$$

 7: **end for**

---

## 2 BACKGROUND

We review the classical notions of Lyapunov stability and the *Lyapunov indirect method* (or *linearization method*), which provide standard tools for analyzing the stability of dynamical systems.

First, we establish a notion of stability of an equilibrium in a dynamical system. Formally, it is characterized with the following definition of Lyapunov stability.

**Definition 2.1** (Lyapunov stability). *Consider the autonomous system $\dot{\boldsymbol{z}}(t) = \boldsymbol{T}(\boldsymbol{z}(t))$, where $\boldsymbol{T} : \mathcal{D} \to \mathbb{R}^d$ is locally Lipschitz with domain $\mathcal{D} \subset \mathbb{R}^d$. Let $\boldsymbol{z}_* \in \mathcal{D}$ be an equilibrium, i.e. $\boldsymbol{T}(\boldsymbol{z}_*) = \boldsymbol{0}$. Then:*

- *$\boldsymbol{z}_*$ is **Lyapunov stable** if for every $\epsilon > 0$ there exists $\delta > 0$ such that $\|\boldsymbol{z}(0) - \boldsymbol{z}_*\| < \delta$ implies $\|\boldsymbol{z}(t) - \boldsymbol{z}_*\| < \epsilon$ for all $t \geq 0$.*

- *$\boldsymbol{z}_*$ is **asymptotically stable** if it is Lyapunov stable and there exists $\delta' > 0$ such that $\|\boldsymbol{z}(0) - \boldsymbol{z}_*\| < \delta'$ implies $\lim_{t\to\infty} \boldsymbol{z}(t) = \boldsymbol{z}_*$.*

- *$\boldsymbol{z}_*$ is **unstable** if it is not Lyapunov stable; i.e., there exists $\epsilon' > 0$ such that for every $\delta > 0$ one can find $\|\boldsymbol{z}(0) - \boldsymbol{z}_*\| < \delta$ but $\|\boldsymbol{z}(t) - \boldsymbol{z}_*\| \geq \epsilon'$ for some $t > 0$.*

Lyapunov's stability establishes the notion of global stability, while the following Hurwitz stability states how the linearized neighborhood near an equilibrium point behaves.

**Definition 2.2** (Hurwitz stability). *Let $\boldsymbol{z}_*$ be an equilibrium of $\dot{\boldsymbol{z}} = \boldsymbol{T}(\boldsymbol{z})$ and let $\boldsymbol{J}(\boldsymbol{z}_*) = \left[\partial T_i / \partial z_j\right]_{\boldsymbol{z}_*}$ denote the Jacobian at $\boldsymbol{z}_*$. Then $\boldsymbol{z}_*$ is **Hurwitz stable** if every eigenvalue $s$ of $\boldsymbol{J}(\boldsymbol{z}_*)$ satisfies $\Re(s) < 0$.*

**Lyapunov's indirect method** gives the following characterization. If $\boldsymbol{J}(\boldsymbol{z}_*)$ is Hurwitz stable, then the equilibrium $\boldsymbol{z}_*$ is *asymptotically stable*. If $\boldsymbol{J}(\boldsymbol{z}_*)$ has an eigenvalue with $\Re(s) > 0$, the equilibrium is *unstable*. When all eigenvalues satisfy $\Re(s) \leq 0$ with some lying on the imaginary axis, the test is inconclusive and higher-order terms must be examined. We direct readers to Khalil (2002) for further details.

While Lyapunov's indirect method addresses continuous-time systems, a parallel criterion applies to discrete dynamics of the form $\boldsymbol{z}_{t+1} = \boldsymbol{T}(\boldsymbol{z}_t)$. In this case, an equilibrium $\boldsymbol{z}_*$ is *Schur stable* if all eigenvalues of $\boldsymbol{J}(\boldsymbol{z}_*)$ lie strictly inside the unit disk, i.e. $|\lambda| < 1$ for every eigenvalue $\lambda$. This condition ensures asymptotic stability. If any eigenvalue satisfies $|\lambda| > 1$, the equilibrium is unstable. Thus, stability analysis for discrete-time optimization algorithms is analogous to the continuous-time setting.

**Notation** We denote the parameter vector by $\boldsymbol{x}$, the first-order momentum by $\boldsymbol{m}$, and the second-order momentum by $\boldsymbol{v}$. The objective function is $f : \mathbb{R}^d \to \mathbb{R}$, defined over the parameter domain. We use $F(\cdot)$ to denote the *implicit objective*, which will be introduced later. Unless stated otherwise,

---

[2]We use the default `AdamW` implementation from PyTorch (PyTorch Contributors, 2025), which computes the denominator as $\sqrt{\hat{v}_t} + e$ rather than $\sqrt{\hat{v}_t + e^2}$. While the alternative will lead to a slightly different implicit objective, our main conclusions remain unaffected.

all operations are elementwise. To avoid ambiguity, we use the following terminology: *equilibria* are stationary points of continuous-time dynamical systems, *fixed points* are update-invariant states of discrete-time algorithms, and *stationary points* are points where the gradient of a function vanishes.

## 3 THE IMPLICIT OBJECTIVE OF ADAMW

We start from the *fixed-point conditions* of AdamW, which characterize the relations that convergence points must satisfy if convergence were to occur. For gradient descent or Adam, these fixed points coincide with the stationary points of the objective $f(\boldsymbol{x})$. In contrast, the decoupled weight decay alters this relationship: equilibria no longer align directly with stationary points of $f(\boldsymbol{x})$ but also depend on the weight decay strength.

Formally, let $[\boldsymbol{x}_t, \boldsymbol{m}_t, \boldsymbol{v}_t]$ denote the AdamW iterates. As $t \to \infty$, bias corrections vanish so that $\hat{\boldsymbol{m}}_t \to \boldsymbol{m}_t$ and $\hat{\boldsymbol{v}}_t \to \boldsymbol{v}_t$. At a fixed point, the updates are stationary, and the iterates satisfy

$$\frac{\boldsymbol{m}}{\sqrt{\boldsymbol{v}} + e} + \lambda \boldsymbol{x} = \boldsymbol{0}, \quad \boldsymbol{m} = \nabla f(\boldsymbol{x}), \quad \boldsymbol{v} = \nabla f(\boldsymbol{x})^{\odot 2}. \tag{1}$$

Eliminating $\boldsymbol{m}$ and $\boldsymbol{v}$ yields the necessary condition

$$\frac{\nabla f(\boldsymbol{x})}{|\nabla f(\boldsymbol{x})| + e} + \lambda \boldsymbol{x} = \boldsymbol{0}. \tag{2}$$

Equation equation 2 exactly describes the fixed-point set of AdamW. These equilibria turn out to coincide with the stationary points of a constrained and regularized optimization problem, which we refer to as the *implicit objective* of AdamW.

**Theorem 3.1.** *Let $f : \mathbb{R}^d \to \mathbb{R}$ be continuously differentiable, and let $\lambda, e > 0$. Then $\boldsymbol{x} \in \mathbb{R}^d$ satisfies the fixed-point condition equation 2 if and only if $\boldsymbol{x}$ is a stationary point of the $\ell_\infty$-constrained and regularized problem*

$$\min_{\boldsymbol{x} \in \mathbb{R}^d} F(\boldsymbol{x}) = f(\boldsymbol{x}) + \tfrac{e}{\lambda} \Phi(\lambda \boldsymbol{x}), \quad \text{subject to} \quad \|\boldsymbol{x}\|_\infty \le \tfrac{1}{\lambda}, \tag{3}$$

*where $\Phi(\boldsymbol{x}) = \sum_{i=1}^d \phi(x_i)$ with*

$$\phi(x) = -|x| - \log(1 - |x|), \quad |x| < 1.$$

*Moreover, since $\phi(x) \to +\infty$ as $|x| \to 1$, the regularizer $\Phi$ serves as a barrier function that naturally enforces the constraint $\|\boldsymbol{x}\|_\infty < \frac{1}{\lambda}$.*

This implicit objective coincides with the Softsign case $\nabla \mathcal{K} = x/(|x| + e)$ of Lion-$\mathcal{K}$ (Appendix A, Chen et al. (2023a)), and it reduces to the pure $\ell_\infty$ constraint of Xie & Li (2024) when $e \to 0$. Additionally, this view of AdamW's implicit objective contrasts with prior interpretations (Loshchilov & Hutter, 2017; Zhou et al., 2024), which describe weight decay as an adaptive $\ell_2$ penalty that depends on optimizer states and therefore deviates from the standard notion of an optimization objective. In contrast, our construction produces a static objective $F(\boldsymbol{x})$ that depends only on the parameters.

### 3.1 FIXED-POINT COINCIDENCE IS INSUFFICIENT

The fact that AdamW's fixed points coincide with the stationary points of equation 3 does not imply that AdamW actually minimizes $F(\boldsymbol{x})$. Beyond this first-order consistency, we require:

*The stable equilibria of AdamW must coincide with the local minima of $F$.*

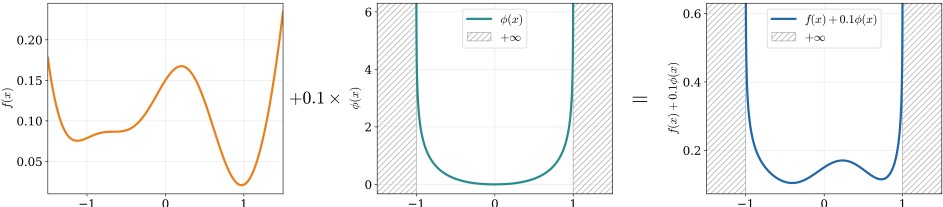

Figure 3: Left: an objective function $f$ that AdamW is applied on. Middle: $\phi$. Right: $f(x) + \frac{e}{\lambda}\phi(\lambda x)$ with $e = 0.1$ and $\lambda = 1$.

For many standard optimization algorithms this alignment holds, often guaranteed by a global Lyapunov function: stable equilibria coincide exactly with local minima. As proved in previous works (Maddison et al., 2018; Chen et al., 2023a), this property is shared across the Lion-$\mathcal{K}$ family, including SGD with momentum, Lion (Chen et al., 2023b), and Muon (Jordan et al., 2024). AdamW, however, is different. A global Lyapunov function *may not exist*, and the equivalence between stable equilibria and local minima holds unconditionally only in the degenerate case $\beta_1 = \beta_2$. When $\beta_1 \neq \beta_2$, as will be shown in the next section, being a local minimum of $F$ is *necessary but not sufficient* for stability.

$$\text{Stable equilibria of AdamW} \subseteq \text{Local minima of } F$$

Consequently, some local minima of $F$ are unstable under AdamW. Since Theorem 3.1 shows that any implicit objective minimized by AdamW must be $F$ (up to composition with a strictly monotone $C^1$ function), this misalignment implies that AdamW may not minimize any general objective function. It also enables explicit counterexamples where the dynamics fail to converge to a fixed point and instead evolve into stable limit cycles.

## 4 STABILITY CONDITIONS AND NON-CONVERGENCE OF ADAMW

We work in the deterministic (batch) gradient setting and analyze AdamW in the continuous-time limit, corresponding to infinitesimal step size. This removes the intervention from gradient noise and learning rate schedules. Importantly, *a sound optimization method should at least converge in this idealized regime*; instability here indicates a fundamental issue in the design. Continuous-time dynamics also require weaker conditions for convergence: gradient flow converges to local minima even without smoothness assumptions (Łojasiewicz, 1984), whereas discrete-time methods such as gradient descent require smoothness and step-size control (Lee et al., 2016). While an analogous discrete-time analysis is possible, the presence of learning rates introduces additional complications that render the resulting expressions far less interpretable; we therefore defer this analysis to Appendix A.4. In practice, learning rates are often small, so continuous-time analysis provides a useful proxy for understanding the behavior of discrete algorithms.

We model the continuous-time dynamics of AdamW by the ODE

$$\dot{\boldsymbol{x}}_t = -\frac{\boldsymbol{m}_t}{\boldsymbol{h}_t + e} - \lambda \boldsymbol{x}_t, \quad \dot{\boldsymbol{m}}_t = a(\nabla f(\boldsymbol{x}_t) - \boldsymbol{m}_t), \quad \dot{\boldsymbol{h}}_t = \frac{b}{2}\left(\frac{\nabla f(\boldsymbol{x}_t)^{\odot 2}}{\boldsymbol{h}_t} - \boldsymbol{h}_t\right). \quad (4)$$

The discrete-time update of AdamW corresponds to a semi-implicit Euler discretization of this system with $\boldsymbol{v}_t = \boldsymbol{h}_t^{\odot 2}$ and the identifications $\eta = \Delta t$, $\beta_1 = 1 - a\Delta t$, $\beta_2 = 1 - b\Delta t$. For large $t$ we adopt the standard approximation $\hat{\boldsymbol{m}}_t \approx \boldsymbol{m}_t$, $\hat{\boldsymbol{v}}_t \approx \boldsymbol{v}_t$.

It is convenient to rewrite the system in vector form (suppressing $t$ for clarity):

$$\frac{\mathrm{d}}{\mathrm{d}t}\boldsymbol{z} = \boldsymbol{T}(\boldsymbol{z}), \quad \boldsymbol{z} := \begin{bmatrix} \boldsymbol{x} \\ \boldsymbol{m} \\ \boldsymbol{h} \end{bmatrix}, \quad \boldsymbol{T}(\boldsymbol{z}) := \begin{bmatrix} -\boldsymbol{m}/(\boldsymbol{h} + e) - \lambda \boldsymbol{x} \\ a(\nabla f(\boldsymbol{x}) - \boldsymbol{m}) \\ \frac{b}{2}(\nabla f(\boldsymbol{x})^{\odot 2}/\boldsymbol{h} - \boldsymbol{h}) \end{bmatrix}. \quad (5)$$

Let $\boldsymbol{z}_* = [\boldsymbol{x}_*, \boldsymbol{m}_*, \boldsymbol{h}_*]$ denote an equilibrium point of the dynamics, that is, a point satisfying $\boldsymbol{T}(\boldsymbol{z}_*) = \boldsymbol{0}$. Differentiating $\boldsymbol{T}$ and evaluating at $\boldsymbol{z}_*$ gives the Jacobian

$$\boldsymbol{J}(\boldsymbol{z}_*) = \begin{bmatrix} -\lambda \boldsymbol{I} & -\mathrm{Diag}(1/(\boldsymbol{h}_* + e)) & \mathrm{Diag}(\boldsymbol{m}_*/(\boldsymbol{h}_* + e)^2) \\ a\nabla^2 f(\boldsymbol{x}_*) & -a\boldsymbol{I} & 0 \\ b\mathrm{Diag}(\nabla f(\boldsymbol{x}_*)/\boldsymbol{h}_*)\nabla^2 f(\boldsymbol{x}_*) & 0 & -b\boldsymbol{I} \end{bmatrix}. \quad (6)$$

### 4.1 LOCAL STABILITY OF ADAMW IN 1D

We first analyze the one-dimensional (1D) case, deferring the higher-dimensional analysis to section 5. Here "1D" refers only to the dimension of $x$, while the full dynamics remain three-dimensional in $[x, m, h]$. As discussed in section 2, the local stability of an equilibrium is determined by the spectrum of $\boldsymbol{J}(\boldsymbol{z}_*)$. To study this spectrum, we use the characteristic polynomial

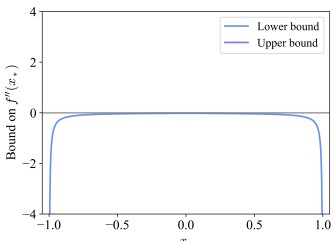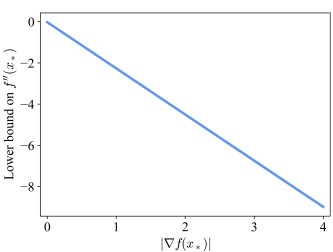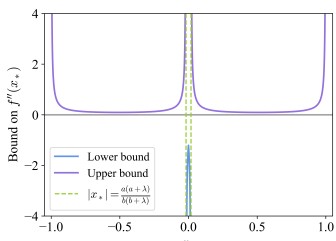

Figure 4: Left: lower bound on $f''(x_*)$ (equation 9) as a function of $x_*$ when $a = 10$, $b = 1$. Middle: the bound as a function of gradient norm $|\nabla f(x_*)|$. Right: case $a = 1$, $b = 10$, where the condition yields either a lower or an upper bound depending on $|x_*|$. We use $\lambda = 1$, $e = 10^{-2}$ in both cases.

$\chi_{\boldsymbol{J}}(s) := \det(sI - \boldsymbol{J}(\boldsymbol{z}_*))$, whose roots coincide with the eigenvalues of $\boldsymbol{J}(\boldsymbol{z}_*)$. In the 1D case, this polynomial simplifies to a cubic:

$$\chi_{\boldsymbol{J}}(s) = s^3 + (a + b + \lambda)s^2 + \left[ ab + a\lambda + \frac{(a-b)H_*}{\hbar_*} + \frac{be\hat{H}_*}{\hbar_*^2} \right] s + \frac{abe\hat{H}_*}{\hbar_*^2}. \qquad (7)$$

Here the coefficients are expressed in terms of the following quantities:

$$\hbar_* := h_* + e, \quad H_* := f''(x_*), \quad \hat{H}_* := F''(x_*),$$

where $F$ is the implicit objective introduced in equation 3. Since $\boldsymbol{z}_*$ is an equilibrium point, we also use the relations

$$h_* = |\nabla f(x_*)| = |m_*|, \quad \hat{H}_* = H_* + \tfrac{\lambda}{e}\hbar_*^2.$$

To analyze the signs of the eigenvalues, we recall this following fact.

**Lemma 4.1** (Routh–Hurwitz criterion for cubics). *All roots of the cubic equation $s^3 + \alpha_2 s^2 + \alpha_1 s + \alpha_0 = 0$ have negative real part (i.e., the system is Hurwitz stable) if and only if*

$$\alpha_2, \ \alpha_1, \ \alpha_0 > 0 \quad \text{and} \quad \alpha_2 \alpha_1 > \alpha_0.$$

**Stable equilibria of AdamW $\subseteq$ Local minima of $F$** We first show that any stable equilibrium of the AdamW dynamics must correspond to a local minimum of the implicit objective $F$.

**Corollary 4.2.** *Assume $a, b, \lambda, e > 0$ and consider the one-dimensional case. If $z_*$ is a Hurwitz stable equilibrium of the AdamW ODE, then it must be a strict local minimum of the implicit objective $F$, i.e. $F''(x_*) > 0$.*

*Proof.* If $\boldsymbol{z}_*$ is Hurwitz stable, then $\alpha_0 = abe\hat{H}_*/\hbar_*^2 > 0$, which implies $\hat{H}_* = F''(x_*) > 0$. □

The subtlety, however, is that $F''(x_*) > 0$ is necessary but *not sufficient*. Even strict local minima of $F$ may correspond to unstable equilibria of AdamW. An additional bound on the Hessian of the original objective $f''(x_*)$ is required.

**Proposition 4.3.** *Assume $a, b, e > 0$ and $\lambda \geq 0$. Let $[x_*, m_*, h_*]$ be an equilibrium of the AdamW ODE in equation 5. Then $[x_*, m_*, h_*]$ is Hurwitz stable if and only if $F''(x_*) > 0$ and*

$$(a + b + \lambda) \left[ ab + a\lambda + \frac{(a-b)f''(x_*)}{\hbar_*} \right] + (b+\lambda)be\frac{F''(x_*)}{\hbar_*^2} > 0, \qquad (8)$$

*which can be rearranged into*

$$(a(a+\lambda) - b(b+\lambda)|\lambda x_*|) f''(x_*) > -\frac{eC}{(1 - |\lambda x_*|)}, \qquad (9)$$

*where $C = (a + b + \lambda)(ab + a\lambda + b\lambda) - ab\lambda > 0$.*

Thus, stability requires two conditions: first, the implicit Hessian must satisfy $F''(x_*) > 0$; second, the inequality in equation 9 must hold, which imposes an additional constraint on the Hessian $f''(x_*)$ of the original loss $f$. We illustrate this constraint in figure 4, showing how the bound behaves for both $a < b$ and $a > b$, and how it depends on the gradient norm $|\nabla f(x_*)|$. In this sense, one may say

*AdamW is a faithful but selective optimizer of $F(x)$.*

Equivalently, AdamW can be interpreted as minimizing $F(x)$, but only under the additional constraint specified by equation 9.

**Remark 4.4** (When the additional condition vanishes). *The extra requirement in equation 9 disappears in two important cases:*

1. ***Adam limit** ($\lambda \to 0$). The implicit objective reduces to $F(x) = f(x)$ as the regularizer and constraint vanishes. The extra requirements becomes $a^2 f''(x_*) > -eC$, which is automatically implied by $F''(x_*) = f''(x_*) \geq 0$. Thus stability reduces to the usual requirement that $x_*$ be a strict local minimum of $f$, explaining that the additional complication arises entirely from the weight-decay term.*

2. ***Equal momentum coefficients** ($a = b$). In this case, equation 8 simplifies into $F''(x_*) > -L$ for $L \geq 0$, which is implied by $F''(x_*) > 0$. Hence stability holds if and only if $x_*$ is a strict local minimum of $F$. (See more in appendix A.8.)*

*In both cases stability reduces to $F''(x_*) > 0$, so the extra bound arises only from weight decay combined with unequal momentum coefficients $a$ and $b$.*

### 4.2 EXAMPLES OF NON-CONVERGENCE

Whenever $a \neq b$, one can construct objectives $f$ that violate the stability condition equation 8 while still satisfying $F''(x_*) > 0$. This creates local minima of $F$ that are stable in the usual optimization sense but unstable under AdamW dynamics. If such a point is the only minimum of $F$ in the domain, the system has no stable fixed point; since trajectories remain bounded, this necessarily yields non-convergent behavior such as limit cycles or other recurrent dynamics. The type of violation depends on the relation between $a$ and $b$: when $a < b$ (equivalently, $\beta_2 < \beta_1$), the condition fails for sufficiently large $f''$, so even strongly convex functions can break stability, which explains this regime is rarely used in practice. By contrast, when $a > b$ (equivalently, $\beta_2 > \beta_1$), which is the regime most common in practice, non-convergence arises only for non-convex objectives. We now provide simple counterexamples illustrating both cases.

**Example 4.5.** *When $\beta_2 < \beta_1$ (so $a < b$), the extra condition in equation 9 imposes an* upper *bound $f''(x_*) \leq U$. This can fail even for simple convex functions. Consider the quadratic*

$$f(x) = (x - 0.5)^2.$$

*With $\lambda = 1$ and $e = 10^{-2}$, the equilibrium is $x_* \approx 0.49$. Choosing $\beta_1 = 0.99$, $\beta_2 = 0.9$, and step size $\eta = 10^{-2}$ gives $a = 1$, $b = 10$. Then equation 9 requires $f''(x_*) < 9.1 \times 10^{-2}$, while in fact $f'' = 2$. Hence the only equilibrium is unstable, and AdamW fails to converge (see figure 1).*

**Example 4.6.** *When $\beta_2 > \beta_1$ (so $a > b$), the condition imposes a* lower *bound $f''(x_*) \geq -L$. This can be violated with a carefully chosen non-convex quartic. Consider*

$$f(x) = 20(x - 0.99)^4 - 0.60(x - 0.99)^2 - 0.099(x - 0.99) + 0.018.$$

*With $a = 10$, $b = 1$ (corresponding to $\beta_1 = 0.90$, $\beta_2 = 0.99$), $\eta = 10^{-2}$, $e = 10^{-3}$, and $\lambda = 1$, the equilibrium at $x_* = 0.99$ has $f''(x_*) \approx -0.60$, violating the required bound $f''(x_*) > -0.22$. Thus the equilibrium is unstable, and AdamW again fails to converge (see figure 2).*

We emphasize that these counterexamples are primarily of theoretical interest, meant to highlight the intricate dynamics of AdamW. In practice, realizing such cases in neural networks would typically require extreme settings, such as unusually large weight decay (e.g., $\lambda > 20$ to drive some $|\lambda x|$ close enough to 1) or deliberately enforcing the atypical regime $\beta_1 > \beta_2$. Neither choice reflects standard training practice. This is unsurprising: modern recipes for network training have been shaped by years of trial and error and by the coevolution of optimizers and architectures, which has naturally filtered out such pathological configurations.

## 5 MULTI-DIMENSIONAL STABILITY OF ADAMW

In the multi-dimensional setting, the analysis becomes more involved and there is no simple stability criterion. We adopt an alternative approach: we transform the linearized system into a third-order ODE and then derive a sufficient condition for convergence using a Lyapunov function, which in turn yields a sufficient condition for stability of equilibria. The linearized dynamics are given by the following lemma.

**Lemma 5.1.** *Let $z_* = (x_*, m_*, h_*)$ be an equilibrium point of the AdamW ODE equation 5. If $a \neq b$, then the linearized dynamics $\dot{z}_t = J(z_*)z_t$ imply that $x_t$ satisfies*

$$\dddot{x}_t + A_2 \ddot{x}_t + A_1 \dot{x}_t + A_0 x_t = 0, \tag{10}$$

*where*

$$A_2 = (a + b + \lambda)I \quad A_0 = ab \left[ \operatorname{Diag}(e/(h_* + e)^2)\nabla^2 f(x_*) + \lambda I \right]$$

$$A_1 = (ab + a\lambda + b\lambda)I + (a - b)\operatorname{Diag}(1/(h_* + e))\nabla^2 f(x_*) + b\operatorname{Diag}(e/(h_* + e)^2)\nabla^2 f(x_*),$$

We can further rewrite equation 10 as a $3d$-dimensional first-order ODE system:

$$\dot{x}_t = y_t, \quad \dot{y}_t = z_t, \quad \dot{z}_t = -A_0 x_t - A_1 y_t - A_2 z_t.$$

To analyze stability, we construct a Lyapunov function of the form

$$V(x, y, z) = \tfrac{1}{2}\left( \|A_2 x + y\|_P^2 + \|A_2 y + z\|_Q^2 + \|y\|_K^2 \right),$$

where $P, Q, K$ are defined as $P = Q A_0 A_2^{-1}$, $K = Q(A_1 - A_0 A_2^{-1})$, and $L = A_2^\top Q(A_1 - A_0 A_2^{-1})$, with $Q$ chosen so that $P, Q, K$ are positive definite. With these definitions, we obtain

$$\frac{\mathrm{d}}{\mathrm{d}t}V(x_t, y_t, z_t) = -\tfrac{1}{2}y^\top(L + L^\top)y.$$

Thus $\dot{V} \leq 0$ whenever $L + L^\top$ is positive definite, leading to the following result.

**Proposition 5.2.** *Consider the system in equation 10. The dynamics are asymptotically stable if there exists a symmetric positive definite matrix $Q$ such that 1. $QA_0$ and $QA_1$ are symmetric, and 2. $QA_0$, $QA_1$, and $Q(A_1 A_2 - A_0)$ are positive definite.*

To complement the multidimensional analysis, we provide a small empirical check on a neural network. We train a three-layer network with hidden dimension 256 and GELU activations (Hendrycks & Gimpel, 2016) on a 5000-sample subset of MNIST, using batch gradients and a continuous-time simulation with Runge–Kutta 4 (step size $2 \times 10^{-3}$, 100k steps). We vary the weight decay $\lambda$ and report the update size $\|\Delta\theta\|_2$ between consecutive steps (Fig. 5, left). For small $\lambda$ ($\lambda = 0$ reducing to Adam or 0.1), updates remain steady and bounded; for large $\lambda$ ($\lambda = 100$), updates fluctuate strongly, indicating non-convergence; for an intermediate $\lambda = 5$, updates quickly decay to machine precision, suggesting convergence.

We further repeat this $\lambda = 5$ setting with 20 random seeds and compute the spectral abscissa of the Jacobian equation 6 at the final equilibrium, i.e. the eigenvalue with largest real part. The histogram in Fig. 5 (right) shows all values strictly negative, indicating that the equilibrium is Hurwitz and consistent with AdamW reaching a stable equilibrium when convergence occurs.

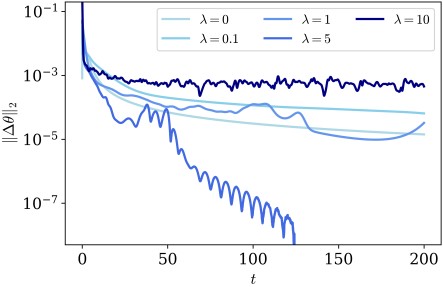
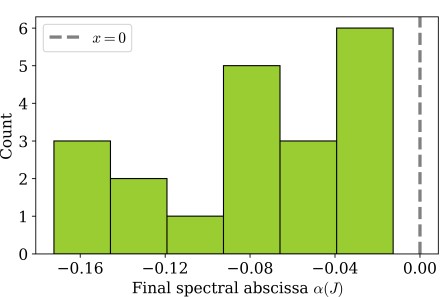

Figure 5: Left: Update size $\|\Delta\theta\|_2$ during training with varying weight decay $\lambda$. Right: Histogram of the spectral abscissa of $J(z_*)$ at the final equilibrium for $\lambda = 5$ over 20 runs.

## 6 RELATED WORKS

**Convergence analysis of Adam.** Adam was introduced by Kingma & Ba (2014) with a convergence proof for convex functions in online convex optimization, but this argument was later shown to contain a gap by Reddi et al. (2019). Their counterexample, however, relies on carefully constructed adversarial online objectives, and thus does not preclude convergence under standard settings (Zhang et al., 2022). Follow-up work established several positive results: De et al. (2018) and Défossez et al. (2020) proved convergence for non-convex functions under bounded gradients and consistent gradient signs, while Zhang et al. (2022) relaxed the bounded-gradient requirement. More recently, Li et al. (2023); Jin et al. (2024) demonstrated convergence under weaker assumptions, narrowing the gap with SGD. A complementary line of work studies Adam's implicit bias and seeks to explain its empirical success (Xie et al., 2024; Vasudeva et al., 2025; Li et al.).

**Adam with decoupled weight decay.** In practice, AdamW (Loshchilov & Hutter, 2017), the decoupled weight decay variant of Adam, has become the default choice over the original Adam. Weight decay itself was introduced by Hanson & Pratt (1988), and while it is equivalent to $L_2$ regularization in SGD, Loshchilov & Hutter (2017) showed this equivalence breaks for adaptive methods and that decoupling improves performance. Since then, the effectiveness of weight decay has been confirmed across architectures and scales (Kosson et al., 2023; D'Angelo et al., 2024; Wang & Aitchison, 2024; Bergsma et al., 2025; He et al., 2025).

**No conflicts with previous studies of AdamW's convergence.** Several works have analyzed AdamW's convergence, but under assumptions that depart from the algorithm as commonly used. Zhou et al. (2024) considered exponentially decaying weight decay $\lambda_k = \lambda(1 - \alpha)^k$, in contrast to the standard constant choice, and defined convergence through $\frac{1}{T}\sum_{k=0}^{T-1} \mathbb{E}[\|\nabla F_k(\boldsymbol{x}_k)\|_2^2]$, where $F_k$ is a state-dependent "dynamically regularized" objective. Recently Li et al. (2025) established a rate of $\mathcal{O}(d^{1/2}T^{-1/4})$ for the criterion $\frac{1}{T}\sum_{k=0}^{T-1} \mathbb{E}[\|\nabla f(\boldsymbol{x}_k)\|_1] \leq \epsilon^2$, but their result requires that the iterates already satisfy the $\ell_\infty$ constraint and assumes small weight decay ($\lambda \leq \mathcal{O}(d^{1/2}T^{-3/4})$) together with large momentum coefficients ($\beta_1 = \mathcal{O}(T^{-1/2})$). Xie & Li (2024) examined AdamW's implicit objective in the limit $e \to 0$, showing convergence to KKT points under suitable learning-rate conditions, but only conditional on AdamW itself converging. In contrast, our results demonstrate that AdamW fails to converge under standard hyperparameters without additional assumptions, and thus are in no conflicts with previous analyses.

## 7 CONCLUSIONS

We studied the dynamics of AdamW and showed that, despite admitting an implicit objective, its equilibria can be unstable, leading to persistent limit cycles on simple problems. Extending the analysis to higher dimensions, we derived sufficient conditions for stability and validated that AdamW converges only to stable equilibria when it converges at all. These results reveal that AdamW's theoretical properties are unexpectedly fragile. Its stability hinges on delicate interactions between curvature, weight decay, and momentum, yet the method remains the default optimizer in practice. This contrast suggests that while AdamW succeeds empirically, its dynamics are theoretically unsatisfying, and points to the need for more natural optimizers that preserve its effectiveness while offering stronger guarantees.

**Limitations** This work provides concrete examples of AdamW's non-convergence and empirical evidence of limit-cycle behavior, but we do not give a formal proof characterizing the dynamics (e.g., cycles versus chaos). In the multi-dimensional case, eigenanalysis is too complex for interpretable necessary-and-sufficient conditions, and we offer only preliminary sufficient ones. Neural network training adds further complications: high dimensionality, discrete learning-rate schedules, stochasticity, and the AdamW's preference of fixed points along manifolds of stationary points, and the expense of large-scale experiments. We regard these as important directions for future study.

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

## A    DEFERRED PROOFS

### A.1    DERIVATION OF THE ADAMW ODE

We now sketch how the AdamW ODE arises as the continuous-time limit of the discrete update, and how the discrete dynamics can be interpreted as a semi-implicit Euler scheme applied to this ODE.

Let $\boldsymbol{g}_t = \nabla f(\boldsymbol{x}_t)$. The AdamW updates are

$$\boldsymbol{m}_t = \beta_1 \boldsymbol{m}_{t-1} + (1-\beta_1)\boldsymbol{g}_t,$$

$$\boldsymbol{v}_t = \beta_2 \boldsymbol{v}_{t-1} + (1-\beta_2)\boldsymbol{g}_t^{\odot 2},$$

$$\boldsymbol{x}_{t+1} = \boldsymbol{x}_t - \eta\left(\frac{\boldsymbol{m}_t}{\sqrt{\boldsymbol{v}_t}+e} + \lambda\boldsymbol{x}_t\right).$$

With the common approximation $\hat{\boldsymbol{m}}_t \approx \boldsymbol{m}_t$, $\hat{\boldsymbol{v}}_t \approx \boldsymbol{v}_t$ and the substitution $\boldsymbol{h}_t = \sqrt{\boldsymbol{v}_t}$, we set

$$\eta = \Delta t, \quad \beta_1 = 1 - a\Delta t, \quad \beta_2 = 1 - b\Delta t.$$

**Continuous-time limit.**    Consider the first-moment update:

$$\frac{\boldsymbol{m}_t - \boldsymbol{m}_{t-1}}{\Delta t} = \frac{-(a\Delta t)\boldsymbol{m}_{t-1} + (a\Delta t)\boldsymbol{g}_t}{\Delta t} = -a\boldsymbol{m}_{t-1} + a\boldsymbol{g}_t.$$

Letting $\Delta t \to 0$, the left-hand side becomes $\dot{\boldsymbol{m}}_t$ and $\boldsymbol{m}_{t-1} \to \boldsymbol{m}_t$, giving $\dot{\boldsymbol{m}}_t = a(\boldsymbol{g}_t - \boldsymbol{m}_t)$. In the same way,

$$\frac{\boldsymbol{x}_{t+1} - \boldsymbol{x}_t}{\Delta t} = -\frac{\boldsymbol{m}_t}{\boldsymbol{h}_t + e} - \lambda\boldsymbol{x}_t \quad \to \quad \dot{\boldsymbol{x}}_t = -\frac{\boldsymbol{m}_t}{\boldsymbol{h}_t + e} - \lambda\boldsymbol{x}_t,$$

and for $\boldsymbol{v}_t = \boldsymbol{h}_t^{\odot 2}$,

$$\frac{\boldsymbol{v}_t - \boldsymbol{v}_{t-1}}{\Delta t} = -b\boldsymbol{v}_{t-1} + b\boldsymbol{g}_t^{\odot 2} \quad \to \quad \dot{\boldsymbol{h}}_t = \tfrac{b}{2}\left(\frac{\boldsymbol{g}_t^{\odot 2}}{\boldsymbol{h}_t} - \boldsymbol{h}_t\right).$$

**Semi-implicit Euler discretization.**    Starting from the AdamW ODE

$$\dot{\boldsymbol{x}}_t = -\frac{\boldsymbol{m}_t}{\boldsymbol{h}_t + e} - \lambda\boldsymbol{x}_t, \qquad \dot{\boldsymbol{m}}_t = a(\nabla f(\boldsymbol{x}_t) - \boldsymbol{m}_t), \qquad \dot{\boldsymbol{h}}_t = \tfrac{b}{2}\left(\frac{\nabla f(\boldsymbol{x}_t)^{\odot 2}}{\boldsymbol{h}_t} - \boldsymbol{h}_t\right),$$

a semi-implicit Euler step of size $\Delta t$ updates $(\boldsymbol{m}, \boldsymbol{v})$ explicitly at $t$ and then advances $\boldsymbol{x}$ using the *new* $(\boldsymbol{m}_{t+1}, \boldsymbol{v}_{t+1})$. Writing $\boldsymbol{v} = \boldsymbol{h}^{\odot 2}$ and $\boldsymbol{g}_t = \nabla f(\boldsymbol{x}_t)$:

$$\boldsymbol{m}_{t+1} = \boldsymbol{m}_t + \Delta t\, a(\boldsymbol{g}_t - \boldsymbol{m}_t) = (1-a\Delta t)\boldsymbol{m}_t + (a\Delta t)\boldsymbol{g}_t,$$

$$\boldsymbol{v}_{t+1} = \boldsymbol{v}_t + \Delta t\, b(\boldsymbol{g}_t^{\odot 2} - \boldsymbol{v}_t) = (1-b\Delta t)\boldsymbol{v}_t + (b\Delta t)\boldsymbol{g}_t^{\odot 2}, \quad \boldsymbol{h}_{t+1} = \sqrt{\boldsymbol{v}_{t+1}},$$

$$\boldsymbol{x}_{t+1} = \boldsymbol{x}_t + \Delta t\left(-\frac{\boldsymbol{m}_{t+1}}{\boldsymbol{h}_{t+1} + e} - \lambda\boldsymbol{x}_t\right) = \boldsymbol{x}_t - \eta\left(\frac{\boldsymbol{m}_{t+1}}{\sqrt{\boldsymbol{v}_{t+1}}+e} + \lambda\boldsymbol{x}_t\right),$$

with the identifications $\eta = \Delta t$, $\beta_1 = 1 - a\Delta t$, $\beta_2 = 1 - b\Delta t$. This is exactly the AdamW update without bias correction, confirming that AdamW implements a semi-implicit Euler discretization of the AdamW ODE.

### A.2    IMPLICIT OBJECTIVE OF ADAMW

**Theorem A.1.** *Let $f : \mathbb{R}^d \to \mathbb{R}$ be continuously differentiable, and let $\lambda, e > 0$. Then $\boldsymbol{x} \in \mathbb{R}^d$ satisfies the fixed-point condition equation 2 if and only if $\boldsymbol{x}$ is a stationary point of the $\ell_\infty$-constrained and regularized problem*

$$\min_{\boldsymbol{x}\in\mathbb{R}^d} F(\boldsymbol{x}) = f(\boldsymbol{x}) + \tfrac{e}{\lambda}\Phi(\lambda\boldsymbol{x}), \quad subject\ to \quad \|\boldsymbol{x}\|_\infty \leq \tfrac{1}{\lambda}, \tag{11}$$

*where $\Phi(\boldsymbol{x}) = \sum_{i=1}^d \phi(x_i)$ with*

$$\phi(x) = -|x| - \log(1 - |x|), \quad |x| < 1.$$

*Moreover, since $\phi(x) \to +\infty$ as $|x| \to 1$, the regularizer $\Phi$ serves as a barrier function that naturally enforces the constraint $\|\boldsymbol{x}\|_\infty < \tfrac{1}{\lambda}$.*

We begin with the following lemma with the regularizer:

**Lemma A.2.** *Let* $\phi : \mathbb{R} \to \mathbb{R} \cup \{+\infty\}$ *be*

$$\phi(x) = \begin{cases} -|x| - \log(1 - |x|), & |x| < 1, \\ +\infty, & |x| \geq 1, \end{cases} \qquad \Phi(\boldsymbol{x}) = \sum_{i=1}^{d} \phi(x_i),$$

*and for* $\lambda > 0$ *define the rescaled barrier* $\phi_\lambda(x) := \lambda^{-1}\phi(\lambda x)$ *and* $\Phi_\lambda(\boldsymbol{x}) := \sum_i \phi_\lambda(x_i)$. *Then* $\phi_\lambda$ *is convex and, for* $|x| < 1/\lambda$,

$$\phi_\lambda'(x) = \frac{\lambda x}{1 - |\lambda x|}, \qquad \phi_\lambda''(x) = \frac{\lambda}{(1 - |\lambda x|)^2}.$$

*Moreover, the conjugate of* $\phi$ *is* $\phi^*(y) = |y| - e \log(|y| + e)$ *with* $(\phi^*)'(y) = \frac{y}{|y|+e}$ *and* $(\phi^*)''(y) = \frac{e}{(|y|+e)^2}$.

*Proof.* The stated derivatives follow by direct differentiation on $(-1, 1)$; convexity holds since $\phi_\lambda'' \geq 0$. The conjugate and its derivatives are obtained by a standard one-dimensional Legendre transform; see the expressions above. $\square$

**Lemma A.3.** *For* $e, \lambda > 0$ *and any* $g, x \in \mathbb{R}$,

$$\frac{g}{|g| + e} + \lambda x = 0 \quad \Longleftrightarrow \quad g + e\,\phi_\lambda'(x) = 0,$$

*and hence the left-hand side is well-defined only if* $|\lambda x| < 1$. *In* $\mathbb{R}^d$, *the equivalence holds element-wise.*

*Proof.* Using Lemma A.2, $\phi_\lambda'(x) = \frac{\lambda x}{1-|\lambda x|}$. If $\frac{g}{|g|+e} + \lambda x = 0$, then $|\lambda x| < 1$ and $\text{sign}(x) = -\text{sign}(g)$. Simple algebra yields $g + e\,\frac{\lambda x}{1-|\lambda x|} = 0$. The converse is identical in reverse. $\square$

**Lemma A.4.** *Let* $F(\boldsymbol{x}) := f(\boldsymbol{x}) + \frac{e}{\lambda}\Phi(\lambda\boldsymbol{x}) = f(\boldsymbol{x}) + e\,\Phi_\lambda(\boldsymbol{x})$. *On* $\{\boldsymbol{x} : \|\lambda\boldsymbol{x}\|_\infty < 1\}$,

$$\nabla F(\boldsymbol{x}) = \nabla f(\boldsymbol{x}) + e\,\nabla\Phi_\lambda(\boldsymbol{x}), \qquad \nabla^2 F(\boldsymbol{x}) = \nabla^2 f(\boldsymbol{x}) + \text{Diag}\left(\frac{\lambda e}{(1 - |\lambda\boldsymbol{x}|)^{\odot 2}}\right).$$

*Proof.* Both identities follow by differentiating coordinatewise and invoking Lemma A.2. $\square$

*Proof of Theorem. Fixed points* $\Longleftrightarrow$ *stationary points.* For AdamW fixed points (discrete or continuous limit without bias corrections), the stationary condition for $\boldsymbol{x}$ reads

$$\frac{\nabla f(\boldsymbol{x}_*)}{|\nabla f(\boldsymbol{x}_*)| + e} + \lambda\boldsymbol{x}_* = 0 \quad \text{(elementwise)}.$$

By Lemma A.3 this is equivalent to

$$\nabla f(\boldsymbol{x}_*) + e\,\nabla\Phi_\lambda(\boldsymbol{x}_*) = \nabla F(\boldsymbol{x}_*) = 0, \quad \|\lambda\boldsymbol{x}_*\|_\infty < 1.$$

Thus fixed points lie in the open cube $\{\|\lambda\boldsymbol{x}\|_\infty < 1\}$ and are precisely the unconstrained stationary points of $F$ there.

Conversely, any stationary point of equation 3 must satisfy $\|\lambda\boldsymbol{x}_*\|_\infty < 1$: since $\Phi(\lambda\boldsymbol{x}) \to +\infty$ as $|\lambda x_i| \uparrow 1$, no optimality condition can be met on the boundary for a continuous $f$. Hence the inequality constraints are inactive, the Lagrange multipliers vanish, and the stationary conditions reduce to $\nabla F(\boldsymbol{x}_*) = 0$, which is equivalent (by Lemma A.3) to the AdamW fixed-point condition. This proves the coincidence of fixed points and stationary points. $\square$

## A.3 STABILITY CONDITION OF ADAMW ODE

Writing the AdamW ODE in a vectorized form, we have:

$$\frac{d}{dt}\boldsymbol{z} = \boldsymbol{f}(\boldsymbol{z}), \quad \boldsymbol{z} := \begin{bmatrix} \boldsymbol{x} \\ \boldsymbol{m} \\ \boldsymbol{h} \end{bmatrix}, \quad \boldsymbol{f}(\boldsymbol{z}) := \begin{bmatrix} -\boldsymbol{m}/(\boldsymbol{h}+e) - \lambda\boldsymbol{x} \\ a(\nabla f(\boldsymbol{x}) - \boldsymbol{m}) \\ \frac{b}{2}(\nabla f(\boldsymbol{x})^{\odot 2}/\boldsymbol{h} - \boldsymbol{h}) \end{bmatrix}. \tag{12}$$

By differentiating the system $\boldsymbol{J}(\boldsymbol{z}) := \partial \boldsymbol{f}(\boldsymbol{z})/\partial\boldsymbol{z}$, we obtain:

$$J(\boldsymbol{z}) = \begin{bmatrix} -\lambda\boldsymbol{I} & -\mathrm{Diag}(1/(\boldsymbol{h}+e)) & \mathrm{Diag}(\boldsymbol{m}/(\boldsymbol{h}+e)^2) \\ a\nabla^2 f(\boldsymbol{x}) & -aI & 0 \\ b\mathrm{Diag}(\nabla f(\boldsymbol{x})/\boldsymbol{h})\nabla^2 f(\boldsymbol{x}) & 0 & -\frac{b}{2}\left(\mathrm{Diag}(\nabla f(\boldsymbol{x})^{\odot 2}/\boldsymbol{h}^2) + I\right) \end{bmatrix}.$$

Noting that at equilibrium point $\boldsymbol{z}_* = [\boldsymbol{x}_*, \boldsymbol{m}_*, \boldsymbol{h}_*]$, we have $\nabla f(\boldsymbol{x})^{\odot 2} = \boldsymbol{h}_*^{\odot 2}$, this yields:

$$\boldsymbol{J}(\boldsymbol{z}_*) = \begin{bmatrix} -\lambda\boldsymbol{I} & -\mathrm{Diag}(1/(\boldsymbol{h}_*+e)) & \mathrm{Diag}(\boldsymbol{m}_*/(\boldsymbol{h}_*+e)^2) \\ a\nabla^2 f(\boldsymbol{x}_*) & -a\boldsymbol{I} & 0 \\ b\mathrm{Diag}(\nabla f(\boldsymbol{x}_*)/\boldsymbol{h}_*)\nabla^2 f(\boldsymbol{x}_*) & 0 & -b\boldsymbol{I} \end{bmatrix}.$$

This following lemma is useful for simplifying the characteristic polynomial:

**Lemma A.5.** *If $[\boldsymbol{x}_*, \boldsymbol{m}_*, \boldsymbol{h}_*]$ is an fixed point of AdamW in Eq. equation 1, then*

$$\nabla^2 F(\boldsymbol{x}_*) = \nabla^2 f(\boldsymbol{x}_*) + \mathrm{Diag}\frac{\lambda e}{(1-|\lambda\boldsymbol{x}_*|)^2} = \nabla^2 f(\boldsymbol{x}_*) + \mathrm{Diag}\frac{\lambda}{e}\left(\boldsymbol{h}_* + e\right)^2.$$

*Proof.* At equilibrium, we have $\frac{\boldsymbol{m}_*}{\boldsymbol{h}_*+e} + \lambda\boldsymbol{x}_* = 0$ and $\boldsymbol{h}_* = |\nabla f(\boldsymbol{x}_*)| = |\boldsymbol{m}_*|$. Hence

$$|\lambda\boldsymbol{x}_*| = \frac{|\boldsymbol{m}_*|}{\boldsymbol{h}_*+e} = \frac{\boldsymbol{h}_*}{\boldsymbol{h}_*+e}.$$

Therefore,

$$\frac{\lambda e}{(1-|\lambda\boldsymbol{x}_*|)^2} = \frac{\lambda e}{(e/(\boldsymbol{h}_*+e))^2} = \frac{\lambda}{e}\left(\boldsymbol{h}_* + e\right)^2.$$

The claim then follows from lemma A.2. $\qquad\square$

**Lemma A.6** (Routh–Hurwitz criterion for cubics). *All roots of the cubic equation $s^3 + \alpha_2 s^2 + \alpha_1 s + \alpha_0 = 0$ have negative real part (i.e., the system is Hurwitz stable) if and only if*

$$\alpha_2,\ \alpha_1,\ \alpha_0 > 0 \quad and \quad \alpha_2\alpha_1 > \alpha_0.$$

*Proof.* ($\Rightarrow$) Suppose all roots satisfy $\Re(s_i) < 0$. Writing the polynomial as

$$(s - s_1)(s - s_2)(s - s_3) = s^3 + \alpha_2 s^2 + \alpha_1 s + \alpha_0,$$

Vieta's relations give

$$\alpha_2 = -(s_1 + s_2 + s_3), \quad \alpha_1 = s_1 s_2 + s_1 s_3 + s_2 s_3, \quad \alpha_0 = -s_1 s_2 s_3.$$

Since $\Re(s_i) < 0$, each coefficient is strictly positive. Moreover,

$$\alpha_2\alpha_1 - \alpha_0 = (-(s_1 + s_2 + s_3))(s_1 s_2 + s_1 s_3 + s_2 s_3) + s_1 s_2 s_3 \geq 0,$$

where non-negativity follows whether the roots are all real or include a complex-conjugate pair.

($\Leftarrow$) Conversely, assume $\alpha_2, \alpha_1, \alpha_0 > 0$ and $\alpha_2\alpha_1 > \alpha_0$. - If $s \in \mathbb{R}$ were positive, then

$$s^3 + \alpha_2 s^2 + \alpha_1 s + \alpha_0 > 0,$$

contradiction. Hence all real roots satisfy $s \leq 0$. Now let $s = u + iv$ with $v \neq 0$. Substituting and separating real and imaginary parts yields

$$3u^2 + 2\alpha_2 u + \alpha_1 = v^2, \qquad u^3 + \alpha_2 u^2 + \alpha_1 u + \alpha_0 = (3u + \alpha_2)v^2.$$

Eliminating $v^2$ gives

$$8u^3 + 8\alpha_2 u^2 + 2(\alpha_2^2 + \alpha_1)u + (\alpha_2\alpha_1 - \alpha_0) = 0.$$

If $u > 0$, every term on the left-hand side is strictly positive, a contradiction. Thus $u \leq 0$, i.e., all complex roots lie in the closed left half-plane.

Therefore, the given conditions are necessary and sufficient for Hurwitz stability. $\qquad\square$

**Corollary A.7.** *Assume $a, b, e > 0$ and $\lambda \geq 0$. Let $[x_*, m_*, h_*]$ be an equilibrium of the AdamW ODE. Then $[x_*, m_*, h_*]$ is Hurwitz stable if and only if $F''(x_*) > 0$ and*

$$[a(a + \lambda) - b(b + \lambda)|\lambda x_*|]f''(x_*) > -\frac{eC}{(1 - |\lambda x_*|)}, \tag{13}$$

*where $C = (a + b + \lambda)(ab + a\lambda + b\lambda) - ab\lambda > 0$.*

*Proof.* Assume $a, b, e > 0$ and $\lambda \geq 0$. Recall from the Routh–Hurwitz criterion (Lemma A.6) that all roots of a cubic $s^3 + \alpha_2 s^2 + \alpha_1 s + \alpha_0$ have negative real part if and only if

$$\alpha_2, \ \alpha_1, \ \alpha_0 > 0 \quad \text{and} \quad \alpha_2 \alpha_1 > \alpha_0.$$

For the characteristic polynomial of $\boldsymbol{J}(\boldsymbol{z}_*)$,

$$\chi_{\boldsymbol{J}}(s) = s^3 + (a + b + \lambda)s^2 + \left[ab + a\lambda + \frac{(a - b)H_*}{\hbar_*} + \frac{beh_*}{\hbar_*^2}\right]s + \frac{abe\hat{H}_*}{\hbar_*^2},$$

we identify

$$\alpha_2 = a + b + \lambda, \quad \alpha_1 = ab + a\lambda + \frac{(a - b)H_*}{\hbar_*} + \frac{beh_*}{\hbar_*^2}, \quad \alpha_0 = \frac{abe\hat{H}_*}{\hbar_*^2}.$$

Since $a, b, e > 0$ and $\lambda \geq 0$, clearly $\alpha_2 > 0$. Moreover, $\alpha_0 > 0$ if and only if $\hat{H}_* = F''(x_*) > 0$, which shows that any Hurwitz stable equilibrium must be a strict local minimum of $F$.

It remains to impose $\alpha_2 \alpha_1 > \alpha_0$. Substituting the definitions and rearranging, one obtains

$$\left[(a + b + \lambda)(a\hbar_* - bh_*) - abe\right]H_* > \left[(a + b + \lambda)(ab + a\lambda + b\lambda) - ab\lambda\right]\frac{e}{1 - |\lambda x_*|}.$$

Using $h_*/\hbar_* = |\lambda x_*|$ and $\hbar_* = e/(1 - |\lambda x_*|)$, this inequality is equivalent to

$$\left[a(a + \lambda) - b(b + \lambda)|\lambda x_*|\right]f''(x_*) > -\frac{eC}{1 - |\lambda x_*|},$$

where

$$C := (a + b + \lambda)(ab + a\lambda + b\lambda) - ab\lambda > 0.$$

Thus the equilibrium $(x_*, m_*, h_*)$ is Hurwitz stable if and only if $F''(x_*) > 0$ and the above inequality holds. $\square$

**Corollary A.8.** *Assume $a, b, e > 0$, and $\lambda \geq 0$. Let $[x_*, m_*, h_*]$ is an equilibrium point of AdamW ODE. If $a = b$, or $\lambda = 0$, then $x_*$ is Hurwitz stable iff $F''(x_*) > 0$.*

*Proof.* By Lemma A.6, Hurwitz stability of an equilibrium $\boldsymbol{z}_* = [x_*, m_*, h_*]$ requires $F''(x_*) > 0$ and

$$(a + b + \lambda)\left[ab + a\lambda + \frac{(a - b)H_*}{\hbar_*} + \frac{beF''(x_*)}{\hbar_*^2}\right] > \frac{abeF''(x_*)}{\hbar_*^2}.$$

This can be rearranged as

$$(a + b + \lambda)\left[ab + a\lambda + \frac{(a - b)H_*}{\hbar_*}\right] + (b + \lambda)\frac{beF''(x_*)}{\hbar_*^2} > 0.$$

If $a = b$, the term involving $(a - b)$ vanishes, and the inequality simplifies to

$$(a + b + \lambda)(ab + a\lambda) + (b + \lambda)\frac{beF''(x_*)}{\hbar_*^2} > 0.$$

Since $a = b > 0$, $\lambda \geq 0$, and $e > 0$, this condition is automatically satisfied whenever $F''(x_*) > 0$.

If $\lambda = 0$, the dynamics reduce to the unconstrained Adam system. In this case $\hbar_* = e$, $h_* = 0$, and $F''(x_*) = H_*$. The inequality then becomes

$$(a + b)\left[ab + \frac{(a - b)H_*}{e}\right] + \frac{b^2 H_*}{e} > 0,$$

which simplifies further to

$$(a + b)ab + \frac{a^2}{e}H_* > 0.$$

Because $a, b, e > 0$, this condition is equivalent to requiring $H_* > 0$, i.e. $F''(x_*) > 0$.

In both cases, stability is therefore equivalent to $F''(x_*) > 0$, completing the proof. $\qquad\square$

### A.4 THE DISCRETE ADAMW SYSTEM

**Definition A.9** (Lyapunov stability for discrete systems). *Consider the autonomous system $z_{t+1} = T(z_t)$, where $T : \mathcal{D} \to \mathbb{R}^d$ with domain $\mathcal{D} \subset \mathbb{R}^d$. Let $z_* \in \mathcal{D}$ be a fixed point, i.e. $T(z_*) = z_*$. Then:*

- *$z_*$ is **Lyapunov stable** if for every $\epsilon > 0$ there exists $\delta > 0$ such that $\|z_0 - z_*\| < \delta$ implies $\|z_t - z_*\| < \epsilon$ for all integers $t \geq 0$.*

- *$z_*$ is **asymptotically stable** if it is Lyapunov stable and there exists $\delta' > 0$ such that $\|z_0 - z_*\| < \delta'$ implies $\lim_{t\to\infty} z_t = z_*$.*

- *$z_*$ is **unstable** if it is not Lyapunov stable; i.e., there exists $\epsilon' > 0$ such that for every $\delta > 0$ one can find $\|z_0 - z_*\| < \delta$ but $\|z_t - z_*\| \geq \epsilon'$ for some integer $t > 0$.*

Lyapunov's stability characterizes robustness of discrete trajectories, while the following Schur stability criterion describes how the linearized neighborhood near a fixed point behaves.

**Definition A.10** (Schur stability). *Let $z_*$ be a fixed point of $z_{t+1} = T(z_t)$ and let $J(z_*) = \left[\partial T_i / \partial z_j\right]_{z_*}$ denote the Jacobian at $z_*$. Then $z_*$ is **Schur stable** if every eigenvalue $\lambda \in \mathbb{C}$ of $J(z_*)$ satisfies $|\lambda| < 1$.*

*Lyapunov's indirect method (discrete form)* states: If $J(z_*)$ is Schur stable, then the fixed point $z_*$ is *asymptotically stable*. If $J(z_*)$ has an eigenvalue with $|\lambda| > 1$, the fixed point is *unstable*.

We first write the standard discrete time AdamW update in a vectorized form:

$$z_{t+1} = T(z_t), \quad z := \begin{bmatrix} x \\ m \\ h \end{bmatrix}, \quad T(z) := \begin{bmatrix} (1 - \lambda\eta)x - \eta\frac{\beta_1 m + (1-\beta_1)\nabla f(x)}{\beta_2' h + (1-\beta_2')\nabla f(x)^{\odot 2}/h + e} \\ \beta_1 m_t + (1 - \beta_2)\nabla f(x) \\ \beta_2' h + (1 - \beta_2')\nabla f(x)^{\odot 2}/h \end{bmatrix}. \quad (14)$$

where $\beta_2' = \frac{1+\beta_2}{2}$. Note that the term corresponding to $x_{t+1}$ involves more terms than the continuous counterpart. This is because AdamW update corresponds to the semi-implicit Euler discretization of the AdamW ODE and we need to write $x_{t+1}$ in terms of $[x_t, m_t, h_t]$ only for analysis. Nevertheless, we will use $m_{t+1}$ and $h_{t+1}$ to simplify expressions in the following. Differentiating this system $J(z) := \partial T(z)/\partial z$, we obtain:

$$J(z_t) = \begin{bmatrix} \frac{\partial x_{t+1}}{\partial x_t} & -\eta\beta_1 \mathrm{Diag}(\frac{1}{h_{t+1}+e}) & \frac{\partial x_{t+1}}{\partial h_t} \\ (1 - \beta_1)\nabla^2 f(x_t) & \beta_1 I & 0 \\ (1 - \beta_2)\mathrm{Diag}(\frac{\nabla f(x_t)}{h_t})\nabla^2 f(x_t) & 0 & (1 + \beta_2)I - \mathrm{Diag}(\frac{h_{t+1}}{h_t}) \end{bmatrix}.$$

where

$$\frac{\partial x_{t+1}}{\partial x_t} = (1 - \lambda\eta)I - \eta(1 - \beta_1)\mathrm{Diag}\left(\frac{1}{h_{t+1} + e}\right)\nabla^2 f(x_t)$$

$$+ \eta(1 - \beta_2)\mathrm{Diag}\left(\frac{m_{t+1}\nabla f(x_t)}{h_t(h_{t+1} + e)^2}\right)\nabla^2 f(x_t)$$

$$\frac{\partial x_{t+1}}{\partial h_t} = \eta\mathrm{Diag}\left(\frac{m_{t+1}}{(h_{t+1} + e)^2}\right)\left(\beta_2' I - (1 - \beta_2')\mathrm{Diag}\left(\frac{\nabla f(x_t)^{\odot 2}}{h_t^2}\right)\right)$$

Similarly, at fixed point $\boldsymbol{z}_* = [\boldsymbol{x}_*, \boldsymbol{m}_*, \boldsymbol{h}_*]$, we have $\nabla f(\boldsymbol{x})^{\odot 2} = \boldsymbol{h}_*^{\odot 2}$, this yields:

$$
J(\boldsymbol{z}_*) = \begin{bmatrix}
\frac{\partial \boldsymbol{x}_{t+1}}{\partial \boldsymbol{x}_t} & -\eta \beta_1 \mathrm{Diag}(\frac{1}{\boldsymbol{h}_*+e}) & \eta \beta_2 \mathrm{Diag}\left(\frac{\boldsymbol{m}_*}{(\boldsymbol{h}_*+e)^2}\right) \\
(1-\beta_1)\nabla^2 f(\boldsymbol{x}_*) & \beta_1 \boldsymbol{I} & 0 \\
(1-\beta_2)\mathrm{Diag}(\frac{\nabla f(\boldsymbol{x}_*)}{\boldsymbol{h}_*})\nabla^2 f(\boldsymbol{x}_*) & 0 & \beta_2 \boldsymbol{I}
\end{bmatrix}.
$$

where

$$
\frac{\partial \boldsymbol{x}_{t+1}}{\partial \boldsymbol{x}_t} = (1-\lambda\eta)\boldsymbol{I} - \eta(1-\beta_1)\mathrm{Diag}\left(\frac{1}{\boldsymbol{h}_*+e}\right)\nabla^2 f(\boldsymbol{x}_*)
$$

$$
+ \eta(1-\beta_2)\mathrm{Diag}\left(\frac{\boldsymbol{h}_*}{(\boldsymbol{h}_*+e)^2}\right)\nabla^2 f(\boldsymbol{x}_*)
$$

Consider the one-dimensional case. The characteristic polynomial of the Jacobian at the fixed point is

$$
\chi_{\boldsymbol{J}}(s) = \det\big(s\boldsymbol{I} - \boldsymbol{J}(\boldsymbol{z}_*)\big) = s^3 + as^2 + bs + c,
$$

where

$$
a = \frac{\eta H_*}{\hbar_*^2}\left[(1-\beta_1)\hbar_* + (\beta_2-1)h_*\right] - (\beta_1+\beta_2+1) + \frac{\eta\lambda}{e}(\hbar_* - h_*),
$$

$$
b = \frac{\eta H_*}{\hbar_*^2}\left[\beta_1\beta_2(\hbar_*-h_*) + \beta_1 h_* - \beta_2 \hbar_*\right] + (\beta_1\beta_2+\beta_1+\beta_2) - \eta\lambda(\beta_1+\beta_2),
$$

$$
c = \beta_1\beta_2(\eta\lambda - 1).
$$

To analyze the stability criterion given the polynomial, we can take advantage of the following lemma.

**Lemma A.11** (Schur–Jury criterion for cubics). *All roots of the cubic polynomial $s^3 + \alpha_2 s^2 + \alpha_1 s + \alpha_0 = 0$ lie strictly inside the unit disk (i.e., the system is Schur stable) if and only if the following four inequalities hold:*

$$
|\alpha_0| < 1, \quad 1 + \alpha_2 + \alpha_1 + \alpha_0 > 0, \quad 1 - \alpha_2 + \alpha_1 - \alpha_0 > 0, \quad 1 - \alpha_1 + \alpha_2\alpha_0 - \alpha_0^2 > 0.
$$

Applying Lemma A.11 reveals the stability condition for the discrete system.

$$
|\beta_1\beta_2(\eta\lambda - 1)| < 1, \quad F''(x_*) > 0,
$$

$$
\frac{\eta}{e}(1-|\lambda x_*|)\,f''(x_*)[(1-|\lambda x_*|)(\beta_1-1)(\beta_2+1) + 2\,|\lambda x_*|(\beta_1-\beta_2)]
$$

$$
+ (\beta_1\beta_2 + \beta_1 + \beta_2 + 1)(2 - \eta\lambda) > 0,
$$

$$
\frac{\eta}{e}(1-|\lambda x_*|)\,f''(x_*)\big[\beta_1\beta_2(\eta\lambda - 1)\big(|\lambda x_*|(\beta_2-1) - (\beta_1-1)\big) + \beta_1\,|\lambda x_*|(\beta_2-1) - \beta_2(\beta_1-1)\big]
$$

$$
- (\beta_1\beta_2 - 1)\big(\beta_1\eta\lambda - \beta_1 + 1\big)\big(\beta_2\eta\lambda - \beta_2 + 1\big) > 0.
$$

A.5 MULTIDIMENSIONAL CASE

**Lemma A.12.** *Let $\boldsymbol{z}_* = (\boldsymbol{x}_*, \boldsymbol{m}_*, \boldsymbol{h}_*)$ be an equilibrium point of the AdamW ODE equation 5. If $a \neq b$, then the linearized dynamics $\dot{\boldsymbol{z}}_t = \boldsymbol{J}(\boldsymbol{z}_*)\boldsymbol{z}_t$ imply that $\boldsymbol{x}_t$ satisfies*

$$
\dddot{\boldsymbol{x}}_t + A_2\ddot{\boldsymbol{x}}_t + \boldsymbol{A}_1\dot{\boldsymbol{x}}_t + \boldsymbol{A}_0\boldsymbol{x}_t = 0, \tag{15}
$$

*where*

$$
\boldsymbol{A}_2 = (a + b + \lambda)\boldsymbol{I}
$$

$$
\boldsymbol{A}_1 = (ab + a\lambda + b\lambda)\boldsymbol{I} + (a-b)\,\mathrm{Diag}(1/(\boldsymbol{h}_*+e))\nabla^2 f(\boldsymbol{x}_*) + b\,\mathrm{Diag}(e/(\boldsymbol{h}_*+e)^2)\nabla^2 f(\boldsymbol{x}_*),
$$

$$
\boldsymbol{A}_0 = ab\left[\mathrm{Diag}(e/(\boldsymbol{h}_*+e)^2)\nabla^2 f(\boldsymbol{x}_*) + \lambda\boldsymbol{I}\right],
$$

*Proof.* Linearize the AdamW ODE equation 5 at the equilibrium $\boldsymbol{z}_* = (\boldsymbol{x}_*, \boldsymbol{m}_*, \boldsymbol{h}_*)$. Let $(\boldsymbol{x}, \boldsymbol{m}, \boldsymbol{h})$ denote infinitesimal perturbations (we drop $\delta$ notation for readability). Writing $H := \nabla^2 f(\boldsymbol{x}_*)$ and

$$U := \mathrm{Diag}\Big(\frac{1}{\boldsymbol{h}_* + e}\Big), \; V := \mathrm{Diag}\Big(\frac{\boldsymbol{m}_*}{(\boldsymbol{h}_* + e)^2}\Big), \; D := \mathrm{Diag}\Big(\frac{\nabla f(\boldsymbol{x}_*)}{\boldsymbol{h}_*}\Big), \; S := \mathrm{Diag}\Big(\frac{e}{(\boldsymbol{h}_* + e)^2}\Big),$$

the linearized dynamics are

$$\dot{\boldsymbol{x}} = -\lambda \boldsymbol{x} - U\boldsymbol{m} + V\boldsymbol{h}, \quad \dot{\boldsymbol{m}} = a(H\boldsymbol{x} - \boldsymbol{m}), \quad \dot{\boldsymbol{h}} = b(DH\boldsymbol{x} - \boldsymbol{h}). \tag{16}$$

Differentiate the first equation and substitute equation 16:

$$\ddot{\boldsymbol{x}} = -\lambda \dot{\boldsymbol{x}} - U\dot{\boldsymbol{m}} + V\dot{\boldsymbol{h}} = -\lambda \dot{\boldsymbol{x}} - aUH\boldsymbol{x} + aU\boldsymbol{m} + bVDH\boldsymbol{x} - bV\boldsymbol{h}.$$

Using the identity

$$VD = \mathrm{Diag}\Big(\frac{\boldsymbol{m}_*}{(\boldsymbol{h}_* + e)^2}\Big)\mathrm{Diag}\Big(\frac{\nabla f(\boldsymbol{x}_*)}{\boldsymbol{h}_*}\Big) = \mathrm{Diag}\Big(\frac{\boldsymbol{h}_*}{(\boldsymbol{h}_* + e)^2}\Big) = U - S,$$

where we used $\boldsymbol{m}_* = \nabla f(\boldsymbol{x}_*)$ and $\boldsymbol{h}_* = |\nabla f(\boldsymbol{x}_*)|$ (elementwise) at equilibrium, we obtain

$$\ddot{\boldsymbol{x}} + \lambda \dot{\boldsymbol{x}} + (a - b)UH\boldsymbol{x} + bSH\boldsymbol{x} = aU\boldsymbol{m} - bV\boldsymbol{h}.$$

Eliminate $U\boldsymbol{m}$ using the first line of equation 16: $U\boldsymbol{m} = -\dot{\boldsymbol{x}} - \lambda \boldsymbol{x} + V\boldsymbol{h}$, and thus

$$\ddot{\boldsymbol{x}} + (\lambda + a)\dot{\boldsymbol{x}} + (a - b)UH\boldsymbol{x} + bSH\boldsymbol{x} + a\lambda \boldsymbol{x} = (a - b)V\boldsymbol{h}. \tag{17}$$

When $a \neq b$, equation 17 gives

$$V\boldsymbol{h} = \frac{1}{a - b}\Big[\ddot{\boldsymbol{x}} + (\lambda + a)\dot{\boldsymbol{x}} + (a - b)UH\boldsymbol{x} + bSH\boldsymbol{x} + a\lambda \boldsymbol{x}\Big]. \tag{18}$$

Differentiate equation 17, substitute $\dot{\boldsymbol{h}} = b(DH\boldsymbol{x} - \boldsymbol{h})$ from equation 16, and use equation 18 to eliminate $V\boldsymbol{h}$:

$$\dddot{\boldsymbol{x}} + (\lambda + a)\ddot{\boldsymbol{x}} + \big[(a - b)UH + bSH + a\lambda \boldsymbol{I}\big]\dot{\boldsymbol{x}} - b(a - b)VD\,H\boldsymbol{x} + b(a - b)V\boldsymbol{h} = 0.$$

Using $VD = U - S$, this becomes

$$\dddot{\boldsymbol{x}} + (\lambda + a)\ddot{\boldsymbol{x}} + \big[(a - b)UH + bSH + a\lambda \boldsymbol{I}\big]\dot{\boldsymbol{x}} - b(a - b)(U - S)H\boldsymbol{x} + b(a - b)V\boldsymbol{h} = 0.$$

Substituting equation 18 to eliminate $V\boldsymbol{h}$ gives

$$\dddot{\boldsymbol{x}} + (a + b + \lambda)\ddot{\boldsymbol{x}} + \Big[(ab + a\lambda + b\lambda)\boldsymbol{I} + (a - b)UH + bSH\Big]\dot{\boldsymbol{x}} + ab\Big[SH + \lambda \boldsymbol{I}\Big]\boldsymbol{x} = 0.$$

Recognizing $U = \mathrm{Diag}(1/(\boldsymbol{h}_* + e))$ and $S = \mathrm{Diag}(e/(\boldsymbol{h}_* + e)^2)$ yields the claimed coefficients

$\boldsymbol{A}_2 = (a + b + \lambda)\boldsymbol{I},$

$\boldsymbol{A}_1 = (ab + a\lambda + b\lambda)\boldsymbol{I} + (a - b)\,\mathrm{Diag}(1/(\boldsymbol{h}_* + e))\nabla^2 f(\boldsymbol{x}_*) + b\,\mathrm{Diag}(e/(\boldsymbol{h}_* + e)^2)\nabla^2 f(\boldsymbol{x}_*),$

$\boldsymbol{A}_0 = ab\,\big[\,\mathrm{Diag}(e/(\boldsymbol{h}_* + e)^2)\nabla^2 f(\boldsymbol{x}_*) + \lambda \boldsymbol{I}\,\big],$

which completes the proof. $\qquad\square$

**Lemma A.13** (First-order reduction). *Let $\boldsymbol{x}_t$ solve the third-order ODE*

$$\dddot{\boldsymbol{x}}_t + \boldsymbol{A}_2\ddot{\boldsymbol{x}}_t + \boldsymbol{A}_1\dot{\boldsymbol{x}}_t + \boldsymbol{A}_0\boldsymbol{x}_t = 0.$$

*Define $\boldsymbol{y}_t := \dot{\boldsymbol{x}}_t$ and $\boldsymbol{z}_t := \dot{\boldsymbol{y}}_t$. Then $(\boldsymbol{x}_t, \boldsymbol{y}_t, \boldsymbol{z}_t)$ satisfies*

$$\dot{\boldsymbol{x}} = \boldsymbol{y}, \quad \dot{\boldsymbol{y}} = \boldsymbol{z}, \quad \dot{\boldsymbol{z}} = -\boldsymbol{A}_0\boldsymbol{x} - \boldsymbol{A}_1\boldsymbol{y} - \boldsymbol{A}_2\boldsymbol{z}.$$

*Proof.* Immediate by substitution: $\dot{\boldsymbol{x}} = \boldsymbol{y}, \dot{\boldsymbol{y}} = \boldsymbol{z}$, and using $\dddot{\boldsymbol{x}} = \dot{\boldsymbol{z}}$ in the original ODE. $\qquad\square$

**Lemma A.14** (Derivative identities)**.** *Let $(\boldsymbol{x}, \boldsymbol{y}, \boldsymbol{z})$ evolve according to Lemma A.13. For any symmetric matrices $\boldsymbol{P}, \boldsymbol{Q}, \boldsymbol{K}$ define*

$$W_1(\boldsymbol{x}, \boldsymbol{y}) := \|\boldsymbol{A}_2\boldsymbol{x} + \boldsymbol{y}\|_{\boldsymbol{P}}^2, \quad W_2(\boldsymbol{y}, \boldsymbol{z}) := \|\boldsymbol{A}_2\boldsymbol{y} + \boldsymbol{z}\|_{\boldsymbol{Q}}^2, \quad W_3(\boldsymbol{y}) := \|\boldsymbol{y}\|_{\boldsymbol{K}}^2,$$

*with $\|u\|_{\boldsymbol{M}}^2 := u^\top \boldsymbol{M} u$. Then*

$$\frac{\mathrm{d}}{\mathrm{d}t} W_1 = 2(\boldsymbol{A}_2\boldsymbol{x} + \boldsymbol{y})^\top \boldsymbol{P}(\boldsymbol{A}_2\boldsymbol{y} + \boldsymbol{z}),$$

$$\frac{\mathrm{d}}{\mathrm{d}t} W_2 = -2(\boldsymbol{A}_2\boldsymbol{y} + \boldsymbol{z})^\top (\boldsymbol{Q}\boldsymbol{A}_0\boldsymbol{x} + \boldsymbol{Q}\boldsymbol{A}_1\boldsymbol{y}),$$

$$\frac{\mathrm{d}}{\mathrm{d}t} W_3 = 2\,\boldsymbol{y}^\top \boldsymbol{K}\boldsymbol{z}.$$

*Proof.* For $W_1$, use $\dot{\boldsymbol{x}} = \boldsymbol{y}$ and $\dot{\boldsymbol{y}} = \boldsymbol{z}$:

$$\frac{\mathrm{d}}{\mathrm{d}t}(\boldsymbol{A}_2\boldsymbol{x} + \boldsymbol{y}) = \boldsymbol{A}_2\boldsymbol{y} + \boldsymbol{z},$$

hence $\frac{\mathrm{d}}{\mathrm{d}t} W_1 = 2(\boldsymbol{A}_2\boldsymbol{x} + \boldsymbol{y})^\top \boldsymbol{P}(\boldsymbol{A}_2\boldsymbol{y} + \boldsymbol{z})$.

For $W_2$, note

$$\frac{\mathrm{d}}{\mathrm{d}t}(\boldsymbol{A}_2\boldsymbol{y} + \boldsymbol{z}) = \boldsymbol{A}_2\boldsymbol{z} + \dot{\boldsymbol{z}} = -\boldsymbol{A}_0\boldsymbol{x} - \boldsymbol{A}_1\boldsymbol{y},$$

so $\frac{\mathrm{d}}{\mathrm{d}t} W_2 = 2(\boldsymbol{A}_2\boldsymbol{y} + \boldsymbol{z})^\top \boldsymbol{Q}(-\boldsymbol{A}_0\boldsymbol{x} - \boldsymbol{A}_1\boldsymbol{y})$.

For $W_3$, $\dot{\boldsymbol{y}} = \boldsymbol{z}$, hence $\frac{\mathrm{d}}{\mathrm{d}t} W_3 = 2\,\boldsymbol{y}^\top \boldsymbol{K}\boldsymbol{z}$. $\qquad\square$

**Lemma A.15** (Lyapunov cancellation)**.** *Let $\boldsymbol{Q}$ be symmetric positive definite and define*

$$\boldsymbol{P} := \boldsymbol{Q}\boldsymbol{A}_0\boldsymbol{A}_2^{-1}, \quad \boldsymbol{K} := \boldsymbol{Q}(\boldsymbol{A}_1 - \boldsymbol{A}_0\boldsymbol{A}_2^{-1}), \quad \boldsymbol{L} := \boldsymbol{A}_2^\top \boldsymbol{Q}(\boldsymbol{A}_1 - \boldsymbol{A}_0\boldsymbol{A}_2^{-1}).$$

*Assume $\boldsymbol{Q}\boldsymbol{A}_0$ and $\boldsymbol{Q}\boldsymbol{A}_1$ are symmetric. Consider*

$$V(\boldsymbol{x}, \boldsymbol{y}, \boldsymbol{z}) := \tfrac{1}{2}\Big(\|\boldsymbol{A}_2\boldsymbol{x} + \boldsymbol{y}\|_{\boldsymbol{P}}^2 + \|\boldsymbol{A}_2\boldsymbol{y} + \boldsymbol{z}\|_{\boldsymbol{Q}}^2 + \|\boldsymbol{y}\|_{\boldsymbol{K}}^2\Big).$$

*Then along trajectories of Lemma A.13,*

$$\frac{\mathrm{d}}{\mathrm{d}t} V(\boldsymbol{x}_t, \boldsymbol{y}_t, \boldsymbol{z}_t) = -\tfrac{1}{2}\,\boldsymbol{y}^\top (\boldsymbol{L} + \boldsymbol{L}^\top)\boldsymbol{y}.$$

*Proof.* Combine Lemma A.14:

$$\frac{\mathrm{d}}{\mathrm{d}t} V = (\boldsymbol{A}_2\boldsymbol{x} + \boldsymbol{y})^\top \boldsymbol{P}(\boldsymbol{A}_2\boldsymbol{y} + \boldsymbol{z}) - (\boldsymbol{A}_2\boldsymbol{y} + \boldsymbol{z})^\top (\boldsymbol{Q}\boldsymbol{A}_0\boldsymbol{x} + \boldsymbol{Q}\boldsymbol{A}_1\boldsymbol{y}) + \boldsymbol{y}^\top \boldsymbol{K}\boldsymbol{z}.$$

With $\boldsymbol{P} = \boldsymbol{Q}\boldsymbol{A}_0\boldsymbol{A}_2^{-1}$ we have $\boldsymbol{Q}\boldsymbol{A}_0 - \boldsymbol{P}\boldsymbol{A}_2 = 0$ and $\boldsymbol{Q}\boldsymbol{A}_1 - \boldsymbol{P} = \boldsymbol{K}$. Substituting yields

$$\begin{aligned}
\frac{\mathrm{d}}{\mathrm{d}t} V &= -(\boldsymbol{A}_2\boldsymbol{y} + \boldsymbol{z})^\top \boldsymbol{K}\boldsymbol{y} + \boldsymbol{y}^\top \boldsymbol{K}\boldsymbol{z} \\
&= -\boldsymbol{y}^\top \boldsymbol{A}_2^\top \boldsymbol{K}\boldsymbol{y} \\
&= -\boldsymbol{y}^\top \boldsymbol{A}_2^\top \boldsymbol{Q}(\boldsymbol{A}_1 - \boldsymbol{A}_0\boldsymbol{A}_2^{-1})\boldsymbol{y} \\
&= -\tfrac{1}{2}\,\boldsymbol{y}^\top (\boldsymbol{L} + \boldsymbol{L}^\top)\boldsymbol{y}.
\end{aligned}$$

$\qquad\square$

**Lemma A.16** (Lyapunov stability for the third-order system)**.** *Consider $\dddot{\boldsymbol{x}}_t + \boldsymbol{A}_2\ddot{\boldsymbol{x}}_t + \boldsymbol{A}_1\dot{\boldsymbol{x}}_t + \boldsymbol{A}_0\boldsymbol{x}_t = 0$. If there exists a symmetric positive definite $\boldsymbol{Q}$ such that*

1. *$\boldsymbol{Q}\boldsymbol{A}_0$ and $\boldsymbol{Q}\boldsymbol{A}_1$ are symmetric;*

2. *$\boldsymbol{Q}\boldsymbol{A}_0$, $\boldsymbol{Q}\boldsymbol{A}_1$, and $\boldsymbol{Q}(\boldsymbol{A}_1\boldsymbol{A}_2 - \boldsymbol{A}_0)$ are positive definite,*

*then the dynamics are asymptotically stable.*

*Proof.* By Lemma A.13, stability of the third-order system is equivalent to stability of its first-order form. Define $\boldsymbol{P}, \boldsymbol{K}, \boldsymbol{L}$ as in Lemma A.15. Assumptions (1)–(2) ensure $\boldsymbol{P}, \boldsymbol{K} \succ 0$ and $\boldsymbol{L} + \boldsymbol{L}^\top \succ 0$. Thus $V$ in Lemma A.15 is positive definite and radially unbounded.

Along trajectories,

$$\frac{\mathrm{d}}{\mathrm{d}t} V(\boldsymbol{x}_t, \boldsymbol{y}_t, \boldsymbol{z}_t) = -\tfrac{1}{2}\, \boldsymbol{y}_t^\top (\boldsymbol{L} + \boldsymbol{L}^\top) \boldsymbol{y}_t \leq 0,$$

so $V$ is nonincreasing. Hence trajectories are bounded and Lyapunov stable.

By LaSalle's invariance principle, the $\Omega$-limit set lies inside $\mathcal{E} = \{(\boldsymbol{x}, \boldsymbol{y}, \boldsymbol{z}) : \boldsymbol{y} = 0\}$. On $\mathcal{E}$, $\dot{\boldsymbol{x}} = 0$, so $\boldsymbol{x}$ is constant; also $\dot{\boldsymbol{y}} = \boldsymbol{z} = 0$. Finally $\dot{\boldsymbol{z}} = -\boldsymbol{A}_0 \boldsymbol{x} = 0$ implies $\boldsymbol{x} = 0$ because $\boldsymbol{Q}\boldsymbol{A}_0 \succ 0$. Thus the only invariant set is $(0, 0, 0)$, so the system is asymptotically stable.

In this derivation we used the fact that $\boldsymbol{A}_2$ is symmetric positive definite and commutes with every matrix. This holds automatically because

$$\boldsymbol{A}_2 = (a + b + \lambda)\boldsymbol{I},$$

a scalar multiple of the identity. □

## B  THE USE OF LARGE LANGUAGE MODELS (LLMS)

We used large language models (LLMs) as a general-purpose assistive tool in preparing this paper. Specifically, LLMs were employed to polish the writing for clarity and readability, and to provide stylistic suggestions. In addition, we interacted with LLMs during the process of generating plots and conducting numerical checks to validate some of the theoretical results. No part of the research questions, theoretical analysis, or main contributions was produced by LLMs; their role was limited to writing assistance and auxiliary support in experimentation and visualization.

