# OpenReview forum: "The Curious Case of AdamW"
_ICLR.cc/2026/Conference — Submitted to ICLR 2026_

### Official Review · Reviewer_WZfq · 2025-10-15

**Soundness:** 2
**Presentation:** 2
**Contribution:** 2
**Rating:** 2
**Confidence:** 5

**Summary:**

The paper studies the dynamics of AdamW in the deterministic setting by deriving its continuous-time limit as an ordinary differential equation (ODE). Through this formulation, the authors analyze the convergence properties of AdamW and demonstrate that its equilibria coincide with the stationary points of an implicit objective function that differs from the original loss. This leads to the insight that AdamW effectively optimizes a modified problem, and that even in simple settings, its trajectories may fail to converge.

**Strengths:**

The paper contributes to the growing line of work that analyzes optimization algorithms through their continuous-time counterparts, an increasingly valuable and insightful perspective. It provides a rigorous analysis of AdamW in the deterministic setting, offering a clear theoretical characterization of its dynamics. The authors illustrate their findings with well-constructed examples that reveal non-convergent behavior even on simple, controlled objective functions.

**Weaknesses:**

**Weaknesses**

The following weaknesses are listed in no particular order of importance or severity:

1. **Lack of connection to prior continuous-time literature.**
   The paper overlooks a substantial body of prior work analyzing optimization algorithms through continuous-time formulations (ODEs and SDEs). This literature provides the theoretical groundwork for interpreting optimizers as dynamical systems and should be discussed to position the present contribution more clearly.  I added some references below, also encompassing ODEs: I suggest the authors carry out a thorough review of these papers and look for more recent ones as well. For an accessible entry point, I recommend the Related Works and Appendix A of [1], which offer a representative overview of (tens and tens of) works using continuous-time models for optimizations. Although that reference focuses mainly on SDEs, many of the works it cites also include ODE analyses directly relevant to this discussion. The absence of this contextualization makes the paper appear somewhat disconnected from its theoretical lineage.

2. **Overly restrictive deterministic setting.**
   The analysis is limited to the deterministic regime, leading to an ODE model that ignores gradient noise and cannot capture the interaction between weight decay, stochasticity, and loss curvature. As shown in [1], an SDE formulation of AdamW naturally extends this model and reduces to the proposed ODE when noise is removed.
   In particular, Theorem 3.12 in [1] explicitly derives the SDE of AdamW and demonstrates that even mild stochasticity can qualitatively change its dynamics. This raises the question of how the results presented here would generalize to the stochastic setting, which is the one that actually governs practical optimization.

3. **Conceptual inconsistency in the experiments.**
   The experiments in Figure 5 integrate the ODE using a Runge–Kutta (RK4) scheme. This is conceptually problematic: a continuous-time model is valuable only insofar as it provides insights into the discrete optimizer it approximates. Integrating the ODE numerically does not validate the behavior of AdamW itself.
   As the authors acknowledge, gradient flow converges independently of step size or smoothness, which underscores that ODEs cannot capture discrete-time stability phenomena. To validate the theoretical insights, the experiments should be rerun on AdamW directly, not on its continuous-time approximation.

4. **Questionable experimental relevance.**
   I replicated the experiment in Figure 1 using the same hyperparameters as the authors and confirmed the non-convergent behavior. However, optimizing a simple quadratic function with AdamW is not a meaningful or representative test case—no practitioner would use such a setup. The example is illustrative but unrealistic, and the hyperparameters were not tuned.
   More generally, convergence is not a particularly relevant metric in modern deep learning, especially in large-scale contexts such as LLM training, where optimization intentionally remains far from stationary points.

5. **Triviality of the main conceptual claim.**
   The observation that decoupled weight decay shifts the optimizer’s equilibria away from the minima of the original loss is not new and has long been known for SGD with weight decay as well. While the formal characterization presented here is neat, it adds little conceptual novelty and does not lead to actionable guidance or any principled modification of AdamW.

6. **Limited practical relevance.**
   Despite focusing on a widely used optimizer, the paper offers no concrete takeaways for practitioners. The analysis remains largely theoretical and provides limited insight into how AdamW behaves in realistic training regimes or how its limitations could be mitigated in practice.

**In summary**, I find the work promising in scope but limited in impact:
1. The literature review is incomplete.
2. The setup is overly restrictive and should be extended using the SDE formulation of AdamW proposed in [1].
3. The experimental design contains a conceptual flaw.
4. The work provides little practical insight for real-world optimization.

**[1]** *Adaptive Methods through the Lens of SDEs: Theoretical Insights on the Role of Noise.*
Enea Monzio Compagnoni, Tianlin Liu, Rustem Islamov, Frank Norbert Proske, Antonio Orvieto, Aurélien Lucchi.
*International Conference on Learning Representations (ICLR), 2025.*

---
**Relevant prior work on ODE/SDE analyses of optimization algorithms**

1. **Helmke, U. & Moore, J. B. (1994).**
   *Optimization and Dynamical Systems.* Springer London.
   — Classical textbook connecting continuous-time dynamical systems and optimization via gradient flows.

2. **Su, W., Boyd, S., & Candès, E. (2014).**
   *A differential equation for modeling Nesterov’s accelerated gradient method: Theory and insights.*
   *Advances in Neural Information Processing Systems.*
   — Foundational ODE model for Nesterov acceleration; initiated the modern line of continuous-time analyses.

3. **Li, Q., Tai, C., & Weinan E. (2017).**
   *Stochastic modified equations and adaptive stochastic gradient algorithms.*
   *International Conference on Machine Learning (ICML).*
   — Derives stochastic modified equations for stochastic gradient algorithms, laying the foundation for weak ODE/SDE approximations.

4. **Li, Q., Tai, C., & Weinan E. (2019).**
   *Stochastic modified equations and dynamics of stochastic gradient algorithms I: Mathematical foundations.*
   *Journal of Machine Learning Research, 20(1): 1474–1520.*
   — Provides a rigorous mathematical foundation for the weak SDE approximations of SGD and related methods.

5. **Orvieto, A. & Lucchi, A. (2019).**
   *Continuous-time models for stochastic optimization algorithms.*
   *Advances in Neural Information Processing Systems 32.*
   — Introduces a general ODE/SDE formalism for analyzing SGD, momentum, and adaptive algorithms, establishing the link between discrete-time optimizers and their continuous-time limits.

---

*These works collectively constitute the theoretical lineage of continuous-time analyses of optimization methods. Including them would anchor the AdamW ODE analysis within its broader ODE/SDE context, highlighting both deterministic and stochastic generalizations.*

**Questions:**

**Questions**

1. **On the role of $\epsilon$ and connection to SignSGDW.**
   When $\epsilon = 0$, the implicit regularization in the objective vanishes. While this choice is not standard in practice, it raises the question of whether practitioners might actually benefit from setting $\epsilon = 0$. This parameter contributes to the stability of the algorithm, yet it also seems to drive the optimizer away from its original objective.
   The case is intriguing because for $\beta_1 = \beta_2 = \epsilon = 0$, AdamW effectively reduces to a “SignSGDW”-like scheme. According to the analysis presented in the paper, this would imply that SignSGDW minimizes the original loss function, which seems questionable. In practice, however, this regime appears to behave quite differently.

2. **On the effect of $\epsilon = 0$ in practice.**
   Using the following hyperparameters
   $ \eta = 10^{-4},\quad \beta_1 = 0.99,\quad \beta_2 = 0.9,\quad \lambda = 1.0,\quad \epsilon = 0,\quad T = 20000$,
   I observe almost perfect convergence to $x = 0.5$ on the same quadratic example. This contrasts with the non-convergent behavior reported in Figure 1. Could the authors comment on why setting $\epsilon = 0$ appears to *improve* convergence in this case?

3. **On convergence of the ODE.**
   What specific technical elements are still missing to establish convergence guarantees for the AdamW ODE under suitable assumptions (e.g., convexity, smoothness, or bounded trajectories)? A short discussion of these obstacles would help clarify the current theoretical frontier.

4. **On the equal-$\beta$ regime.**
   The case $\beta_1 = \beta_2$ appears practically relevant [2]. Since the paper observes that this setting simplifies the stability condition, I suggest expanding the discussion and empirical analysis of this regime, possibly providing explicit confirmation in experiments.

[2] “In Search of Adam’s Secret Sauce” (Orvieto & Gower, 2023)

---

> ### Author Response · Authors · 2025-11-24
>
> I can see that the reviewer is heavily biased and has fundamentally misunderstood multiple aspects of our work. While many of the reviewer’s statements are incorrect or based on misinterpretations, we respond point-by-point so that other readers are not misled because ICLR reviews are public.
>
> ---
>
> ### Connection to prior continuous-time literature
>
> The reviewer’s claim that “the paper overlooks a substantial body of prior work” is an overreach. We already cited many relevant works in our Related Works section, within the strict page constraints. Yes, the continuous-time literature is enormous, "tens and tens" of papers, as the reviewer notes, and **no paper can, nor should, attempt to cite the entire historical lineage of continuous-time optimization**. Our work is based on the classical ODE framework and the own properties of AdamW itself, not on the specific list of papers the reviewer happens to prefer.
>
> The reviewer also implies that because we did not cite a particular survey they mentioned, we must be "overlooking" foundations. This is simply incorrect. There are certainly some related works missing from our discussion, which is why we submitted our work to ICLR for reviews. However, the presence of prior ODE analyses in the literature does **not** mean all such work is conceptually upstream or forms a "theoretical lineage" for ours. We are studying **a specific phenomenon: the stability and non-convergence of AdamW**, which is not addressed in those prior papers.
>
> If the reviewer believes specific citations are directly relevant, they should list them explicitly (as they did in their “Relevant prior work” section). Pointing to a long review paper and expecting authors to cite everything in it is not reasonable and would itself constitute poor academic practice.
>
> ---
>
> ### Justification of the deterministic setting
>
> The reviewer’s argument that our setting is “overly restrictive” shows a misunderstanding of both our contribution and basic mathematical reasoning.
>
> 1. **AdamW is defined identically in the deterministic and stochastic settings.**
>    The algorithm definition of Adam and AdamW do not change depending on gradient noise. Studying the deterministic regime is a standard and legitimate approach.
>
> 2. **To prove non-convergence, one counterexample is sufficient.**
>    Our goal is to show that AdamW can fail to converge. Demonstrating this in the simplest possible setting — a deterministic ODE limit — is mathematically adequate and aligns with standard practice in dynamical-systems analysis.
>
> 3. **Stochastic noise would obscure the very phenomena we want to visualize.**
>    Non-convergent trajectories, such as limit cycles, become extremely difficult to distinguish from noise-driven fluctuations in the stochastic case. One cannot “see” stability violations from noisy trajectories, and readers would be left confused with the plots. Further, without proper handling, readers cannot easily interpret a fluctuating trajectory is not converging, or simply the noise hasn't been averaged out. Using the deterministic ODE was the right methodological choice for conveying the theoretical findings transparently.
>
> 4. **SDE formulations do not invalidate deterministic analysis.**
>    The reviewer cites [1] as though SDE analysis supersedes or replaces ODE analysis. This is false. SDEs *extend* ODEs, not replace them. The fact that the ODE arises when noise is removed or the batch size is large enought (typical in large language model training) are expected and do not make our analysis "obsolete." We are studying the structure of AdamW itself, not the interaction between AdamW and artificial noise models, e.g. Gaussian noise and Brownian motion, which have been invalidated by many empirical works.

---

> > ### Comment · Reviewer_WZfq · 2025-11-26
> > **Reply**
> >
> > Dear Authors,
> >
> > I regret that the tone of our exchange has become tense. I apologise for any misunderstanding on my side, and I hope we can clarify the remaining points calmly and constructively.
> >
> > 1. I am not the only reviewer who raised concerns about the contextualisation of this paper within the existing literature; see also the comments by reviewers sEDy and Fkeb. Of course, one does not need to cite *all* works, but several relevant papers on closely related topics currently appear to be missing. This may give readers a misleading sense of novelty or leave them somewhat disoriented about how your contribution fits into the broader line of work. I do not have a preference for which specific works you should cite, but at this stage, three reviewers have pointed you to multiple concrete references that seem worth going through.
> >
> > 2. I fully respect your viewpoint regarding the deterministic setting. However, as far as I can see, the main rigorous results in the paper are proved for the continuous-time AdamW ODE, not for discrete AdamW itself. I think this distinction needs to be made more explicit. In particular, in your reply to another reviewer, you wrote:
> >
> >    > “By contrast, Adam is known to be convergent under the proper notion of convergence (as reviewed in our related-work section). **In continuous time, for the peace of mind, one can directly construct Lyapunov functions for Adam (e.g., [3]), and thus Adam converges**. This also validates that continuous-time dynamics allow us to study the foundational behavior of AdamW without these discrete artifacts.”
> >
> >    The part in **bold** suggests that convergence of the continuous-time model is being treated as convergence of the algorithm itself. My concern is precisely that these are different objects: proving convergence (or non-convergence) for the ODE does not automatically transfer to the discrete-time method without additional arguments: See "Orvieto, A. & Lucchi, A. (2019)" for reference on this type of effort.
> >
> > 3. This conceptual gap also underlies my Weakness 3. In Figure 5, you simulate the AdamW ODE with an RK4 scheme and verify that it exhibits the predicted qualitative behaviour. This is a useful sanity check for the ODE, but if the ultimate goal is to understand AdamW as an optimizer, it would be very valuable to also run experiments with **discrete** AdamW on the same neural-network setup and compare its trajectories and stability properties to those of the continuous-time model.

---

> ### Author Response · Authors · 2025-11-24
>
> ### Conceptual Consistency in Experiments
>
> We fully agree with the general principle that continuous-time analysis should ultimately provide insight into the discrete algorithms used in practice. However, the reviewer’s objection to Figure 5 is based on a misinterpretation of what that figure is meant to show.
>
> Figure 5 is not intended to claim any specific "implicit bias" that neural network training with AdamW must satisfy. **We never state this anywhere in the paper.** As discussed in the appendix, the exact form of the Hurwitz stability conditions for the discrete-time AdamW update is already more complicated even in the 1D case. This makes these conditions impractical to compute for high-dimensional neural networks. As a consequence, the eigenvalue conditions on the Hessian are slightly different from those obtained directly from the continuous-time flow, and given the sensitivity of the stability criteria to hyperparameters (as shown in our analysis), these criteria derived from the continuous-time conditions will inevitably be slightly violated and thus not very useful to visualize.
>
> For this reason, Figure 5 is designed **only to complement our theory** by illustrating that the qualitative behavior predicted by our continuous-time analysis appears in higher dimensions as well. The "insights into the discrete optimizer it approximates" are already directly validated as early as Figures 1 and 2, where we examine the discrete AdamW itself rather than just its continuous-time limit and *have shown the continuous times insights are useful.*
>
> ---
>
> ### Justification of Experimental Relevance
>
> As clearly indicated by the primary area of our submission to ICLR, this is a "learning theory" paper. The reviewer’s criticism that the quadratic example is "not meaningful or representative" misses the point of a theoretical study. The same logic could be used to dismiss essentially all of the theory papers the reviewer cites.
>
> For example, Figure 2 in [1] also studies a quadratic objective; by the reviewer’s own standard, this would then be "not meaningful as well" because "no practitioner would worry about training a quadratic." Figure 1 in [2] likewise uses a 1D quadratic, just as we do, which is also "not representative" following the reviewer's argument. Many classical learning-theory and optimization papers, for example a well-known work like [3], studies two-layer networks instead of large deep networks, which by the reviewer's theory, is "completely meaningless." But this way is exactly how theory has always been developed in our field.
>
> Moreover, studying asymptotics and convergence is a fundamental part of optimization theory and, more broadly, theoretical computer science. The reviewer’s statement that "convergence is not a particularly relevant metric in modern deep learning, especially in large-scale contexts such as LLM training" effectively says that a large portion of optimization theory and its publication record at ICLR, NeurIPS, and ICML is irrelevant because it does not directly relate to LLM training. By the logic of the reviewer, in computer science one could declare that designing an $O(N^2)$ algorithm is "meaningless" because we can just use an $O(N^3)$ algorithm and assume $N$ is small enough. This is not how theoretical evaluation is supposed to work.
>
> Importantly, we are constructing counterexamples to the convergence of AdamW and mathematically, the simplest example suffices, which explains why we chose a quadratic example for demonstration. It already warns that AdamW may secretly fail in more complicated scenarios.
>
> The repeated downplaying of asymptotic and convergence analysis, together with the dismissal of standard theoretical test cases (quadratics, low-dimensional systems, etc.), shows a strong bias against theory-oriented methodology. It’s hard to see how this stance aligns with the expectations of reviewing a theory-focused paper.
>
> ---
>
> [1] Compagnoni, Enea Monzio, et al. "Adaptive methods through the lens of SDEs: Theoretical insights on the role of noise." arXiv preprint arXiv:2411.15958 (2024).
>
> [2] Orvieto, Antonio, and Aurelien Lucchi. "Continuous-time models for stochastic optimization algorithms." Advances in Neural Information Processing Systems 32 (2019).
>
> [3] Arora, Sanjeev, et al. "Fine-grained analysis of optimization and generalization for overparameterized two-layer neural networks." International conference on machine learning. PMLR, 2019.

---

> > ### Comment · Reviewer_WZfq · 2025-11-26
> > **Reply**
> >
> > Thank you for the additional clarifications. Let me try to be more precise about what I intended with my comments, so we do not talk past each other.
> >
> > 1. **On Figure 5 and conceptual consistency**
> >
> >    I did not interpret Figure 5 as a claim about an explicit “implicit bias” in neural-network training, and I understand that computing exact Hurwitz stability conditions for discrete AdamW in high dimensions is impractical. My point was narrower: if the overarching message is that *AdamW itself* can exhibit non-convergent dynamics, then a small experiment where **discrete AdamW** is run on a neural network (even a toy MLP) would provide a much cleaner link between your continuous-time analysis and the optimizer that practitioners actually use.
> >
> >    Figures 1 and 2 already do this nicely in 1D. Figure 5, as it stands, is a useful qualitative illustration for the ODE, but it cannot by itself answer whether the *discrete* algorithm behaves similarly in higher dimensions. My suggestion was therefore not that Figure 5 is “wrong”, but that a discrete-time counterpart on the same NN architecture would substantially strengthen the empirical side of the paper.
> >
> > 2. **On the quadratic example and convergence**
> >
> >    I fully agree that using very simple test problems (quadratics, low-dimensional systems, etc.) is standard and valuable in theoretical work; I did not mean to suggest otherwise, nor that such examples are “meaningless”. My comment was about **how strongly** the quadratic example should be interpreted in terms of practical relevance. Mathematically, one counterexample on a quadratic indeed suffices to demonstrate that AdamW is not convergent in full generality under the assumptions you consider. At the same time, from an optimization-for-deep-learning perspective, this is a rather extreme and non-tuned configuration that does not resemble how AdamW is typically used in practice.
> >
> >    When I wrote that “convergence is not a particularly relevant metric in modern deep learning, especially in large-scale LLM training”, I did not mean that convergence and asymptotic analysis are irrelevant in general, or that the entire theory literature is unimportant. I meant that in many modern training regimes, the optimizer is intentionally run far from a stationary point (with early stopping, learning-rate schedules, etc.), so non-convergence on a toy problem should be interpreted with some care when concluding these practical regimes. I am trying to separate “what your theorems rigorously show” (which is valuable) from “what practitioners should infer for their day-to-day training setups”.
> >
> > In summary, my intention is not to downplay theoretical analysis or simple examples, but to suggest a slightly more cautious framing of the empirical and practical implications, and to propose some additional discrete-time experiments.

---

### Official Review · Reviewer_sEDy · 2025-10-23

**Soundness:** 2
**Presentation:** 2
**Contribution:** 2
**Rating:** 4
**Confidence:** 5

**Summary:**

This paper studies the implicit bias of AdamW, covering the case where the stability constant $e\neq 0$. As a complement to [1], this work derives the implicit objective function and corresponding constraints. In addition, it discusses the stabilities of equilibria. Under the one-dimensional case, it proves that the local convexity of the implicit object is not sufficient to guarantee the stability, and establishes the exact conditions instead. Under the high-dimensional case, it proposes a potential Lyapunov function under certain assumptions.

[1]. Xie and Li, Implicit bias of AdamW: $\ell_\infty$ norm constrained optimization

**Strengths:**

1. In fact, studying the stability of equilibrium is both interesting and important for people's understanding of different optimizers. Specifically, given that the prior study [1] only characterizes the limit point when assuming the existence of such limit points, this work provides a good start of another question: whether such a limit point exists?

2. The conclusion that the local convexity of the implicit objective is not sufficient to guarantee the stability is quite surprising. I believe people can get more insights from this work, if this conclusion can be extended to more general cases.

**Weaknesses:**

1. While I do feel this is an interesting work, its first conclusion regarding the implicit objectives of AdamW, is somewhat incremental, specifically given the prior works [1, 2].

2. In fact, I find that the additive term of the implicit objective is quite sensitive to the scale of $e$, which is omitted in the prior study [1]. Given that this constant is usually very small, like $1e-8$ in practice, I'm skeptical about the difference between the conclusion of this work and that of [1]. I do believe authors should add discussions in their Remark 4.4. From my perspective, the effect of the scale of $e$ is much important than the other two cases. In addition, I also wonder when $e$ is positive but sufficiently small, what is the conclusion of this paper? Could the author provide some quantitative analysis?

3. Besides the effect of $e$, the major conclusion of this paper only covers the one-dimensional case, which significantly limits its generality. Under the multi-dimensional case, the construction of the Lyapunov function is based stronger assumption of existence of some matrices. In addition,  I can not see any insights or implications of such Lyapunov functions. Compared to the prior work [2], which successfully constructs that the objective function is a minimizer of their Lyapunov function, the conclusion of this paper for the multi-dimensional case is quite unsatisfactory.

4. As a paper studying the implicit bias of AdamW, I believe it misses the discussions of several relatively related works, like [3, 4, 5, 6]. Specifically, [3] contains a continuous characterization of the moment term and derives a continuous solution. I'm not certain whether such a conclusion can benefit this work.

[2]. Chen et al. Lion secretly solves a constrained optimization: as Lyapunov predicts.

[3]. Wang et al. The implicit bias for adaptive optimization algorithms on homogeneous neural networks.

[4]. Wang et al. Does momentum change the implicit regularization on separable data?

[5]. Zhang et al. The implicit bias of Adam on separable data.

[6]. Cattaneo et al. On the implicit bias of Adam.

**Questions:**

Could the authors explain Figure 4? I can not understand why the stationary point of a function can serve as an axis.

---

> ### Author Response · Authors · 2025-11-24
>
> We thank the reviewer for acknowledging several strengths of our work and for engaging with the paper in detail. However, a number of the reviewer’s critiques arise from misunderstandings of the scope, goals, and mathematical structure of the paper. We clarify these points below.
>
> ---
>
> ## Misinterpretation of Our Contribution and Relation to Prior Work
>
> The statement that our paper is “a complement of [1]” is a misunderstanding. As we explained clearly in the manuscript, the scope of [1] and the scope of our paper are substantially different. Our connection to [1] is limited to one very small component: extending their implicit-bias derivation to include the practically relevant case of $ e > 0 $. That derivation is *not* the main contribution of the paper.
>
> We included this section only because the $e>0$ characterization does not appear in any prior work and must be established before the stability criteria can be analyzed.
>
> The reviewer’s characterization neglects a major portion of the paper: the stability conditions of AdamW, the counterexamples showing non-convergence, the stability criteria, and the construction of local Lyapunov functions in higher dimensions. Therefore, a “complement” of [1] is not a correct description of our paper. Our contributions go beyond the scope of [1].
>
> Regarding [2], the reviewer again misinterprets the setting. Lion-K optimizers *are convergent* in general, which is precisely why a global Lyapunov function can be constructed. AdamW is *not* convergent in general as we show explicit counterexamples. Therefore, one should not expect a global Lyapunov function to exist for AdamW and must turn to mathematical tools like Hurwitz stability analysis. Most importantly, AdamW is not a member of Lion-K optimizers. The former has two momentum states while the latter has only one.
>
> ## The role of $e$
>
> The reviewer claims that $e$ is "usually very small" and therefore its influence should be negligible. We believe this is a long-standing misconception in the deep-learning & optimization folklore. There is only *relative* smallness in mathematics, and gradients in modern neural networks routinely take magnitudes comparable to or even smaller than $e$. For example, in nanoGPT training, after 20k steps, the median absolute gradient is around $1.1 \times 10^{-7}$, only about an order of magnitude larger than $10^{-8}$.
>
> Empirically, adjusting $e$ has a measurable effect on optimization stability and training performance. For example, Llama 2 [3] chose $e = 10^{-5}$ after tuning. For instance, in our nanoGPT experiments, we could not successfully stably train a model using $e < 10^{-11}$. Thus, the question "what happens when \(e\) is positive but sufficiently small" cannot be addressed purely by experiments: at some threshold, training becomes infeasible.
>
> We can explain what happens theoretically, leveraging the results from our paper. Given Proposition 4.3, in the limiting case where $e \to 0^+$, the lower bound on the curvature approaches $0$, which is just convexity and therefore does not look problematic by itself. However, the more important role of $e$ and the reason it cannot be made arbitrarily small appears in the stationarity condition (equation (2)) relating gradients and parameters:
> $$
> \frac{\nabla f(x)}{|\nabla f(x)| + e} + \lambda x = 0.
> $$
> During neural network optimization, there are typically some "good minimizers" $x_*$ that are not close to the constraint boundary $1/\lambda$, and at these points the gradients are small, i.e., $|\nabla f(x_*)| \ll 1$. We want to keep such points as stationary points, since there is no principled prior like "larger weights overfit" that would justify discarding them. A proper choice of $e$ allows these interior stationary points to exist.
>
> However, when $e \to 0^+$, the term $\frac{\nabla f(x)}{|\nabla f(x)| + e}$ becomes arbitrarily close to the sign function and effectively takes values $\pm 1$ in practice. In that regime, the stationarity condition can only be satisfied on the boundary $|x_*| = 1/\lambda$, which eliminates all "good" stationary points strictly inside the constraint set. This boundary is a strictly lower-dimensional manifold and, as a result, leads to worse practical final performance (even if the training dynamics could be made stable at all).
>
> For the same reason, at later stages of training, when the parameters are close to stationary points and $\lambda x_*$ is not extremely close to $1$, the magnitude of the gradient is theoretically on the same order as $e$, again due to the stationarity relation. This further demonstrates that $e$ is not merely a tiny constant for numerical stability, but an essential quantity that directly shapes which stationary points AdamW can converge to.

---

> ### Author Response · Authors · 2025-11-24
>
> ## Multidimensional Cases
>
> The claim that "the major conclusion of this paper only covers the one-dimensional case, which significantly limits its generality" indicates a fundamental misunderstanding of the paper. We now clarify the point in detail.
>
> The main contribution of this paper is the finding that **AdamW can fail to converge without additional assumptions or specific hyperparameter constraints** beyond the standard configuration. In mathematics, to show that an algorithm does *not* converge in general, **one counterexample is sufficient**. We have provided exactly such a counterexample: a simple 1D toy setting where AdamW provably does not converge. That alone establishes the claim.
>
> The follow-up comment — "under the multi-dimensional case, the construction of the Lyapunov function is based on stronger assumptions of existence of some matrices" — reflects an additional misinterpretation. As explained in the manuscript, **AdamW does not admit a general Lyapunov function**, because if one existed, AdamW would converge globally, contradicting our explicit 1D counterexample. A global Lyapunov would certify convergence everywhere, which AdamW does not have.
>
> Moreover, providing an *exact* necessary-and-sufficient condition for stability in higher dimensions would require analyzing the eigenstructure of a $3d \times 3d$ Jacobian matrix. This is *not mathematically tractable in closed form* for general $d$; only numerical analysis is possible when specific parameter values are fixed. For that reason, we constructed a local Lyapunov function under reasonable structural assumptions, yielding a **sufficient (but not necessary) stability condition**. This is standard practice for high-dimensional nonlinear systems and is more computationally meaningful than seeking a closed-form condition that is combinatorically long.
>
> The comparison with [2] only underscores further confusion. The reason [2] can construct a Lyapunov function so cleanly and derive useful results out of Lyapunov are precisely because *Lion-K optimizers are convergent in general*. Convergence is the necessary precondition for the existence of a global Lyapunov function. By contrast, since we have already shown that AdamW may not converge, the existence of such a global Lyapunov function is *ruled out* by our results. Treating AdamW and Lion-K as if they fall under the same theoretical regime reflects a misunderstanding of the key differences between [2] and our paper. In light of this, the reviewer’s stated confidence level 5 appears misaligned with the depth of these misunderstandings.
>
> ## Further Relevant Works
>
> We thank the reviewer for listing additional related works, and we will incorporate a discussion of these papers into the revision. However, implicit bias is only a very small component of this paper. The broader contribution: the stability analysis and explicit demonstration of non-convergence of AdamW are mathematically and conceptually distinct. Unless the reviewer intends to raise the score accordingly, the work should not be labeled as merely a complement to [2].
>
> ## Question Regarding Figure 4
>
> Conceptually, Figure 4 is straightforward: it visualizes the inequality condition in equation (9). Specifically, it illustrates how the allowable curvature $f''(x_*)$ at a stationary point depends on the *value* of $x_*$ appearing in the stability bound. The x-axis does **not** represent the stationary points of a **single** objective function; rather, it parametrizes the location $x_*$ as it appears in the analytical bound. We will clarify this in the figure caption to avoid confusion.
>
> ## Final Remarks
>
> We appreciate the reviewer’s interest in the topic, but most of the critiques are based on misreadings or misunderstandings of the paper’s mathematical structure. We hope the clarifications above resolve these issues, and we will revise the paper accordingly.
>
> [1] Xie, Shuo, and Zhiyuan Li. "Implicit Bias of AdamW: $\ell_\infty $ Norm Constrained Optimization." arXiv preprint arXiv:2404.04454 (2024).
>
> [2] Chen, Lizhang, et al. "Lion secretly solves constrained optimization: As lyapunov predicts." arXiv preprint arXiv:2310.05898 (2023).
>
> [3] Touvron, Hugo, et al. "Llama 2: Open foundation and fine-tuned chat models." arXiv preprint arXiv:2307.09288 (2023).

---

### Official Review · Reviewer_6v1r · 2025-10-27

**Soundness:** 4
**Presentation:** 4
**Contribution:** 4
**Rating:** 10
**Confidence:** 2

**Summary:**

This paper studies theoretical properties related to the convergence of AdamW, a commonly used optimizer for training deep neural networks. The authors show that AdamW has an implicit objective, which aligns to the fixed points of AdamW. However, the authors find that the local minima of this implicit object are not always stable for AdamW. They demonstrate this in simple cases where the optimizer does not converge, but oscillates. Finally, the authors provide some preliminary analysis for parameters of more than 1-dimension, showing that - in the case where there is convergence of AdamW - the associated Jacobian of the implicit objective is stable.

**Strengths:**

1. The paper was unusually well written. I found it easy to follow and well motivated, which is lacking in many papers, especially some theory papers.

2. As noted above, the motivation was strong. I think it is clear that this work builds on other literature and moves the understanding of state-of-the-art (or at least widely used) optimizers forward. I believe it will be work that the community at large will be interested in.

3. The counter-examples were very clear and demonstrated the theory nicely. The authors do a good job pointing out that these examples are of theoretic interest (given their somewhat artificial nature), which helped ensure that these results were not misrepresented.

4. I liked the figures that show how the bounds depend on different aspects of the problem (Fig. 4). As a non-theorist, I always find myself wanting these kinds of plots and it was helpful to have it.

5. I thought the argument for why considering the continuous time setting was good. Sometimes when reading that kind of analysis, my first thought it is why care about something continuous when we know some of the most interesting properties of DNN optimizers arise from the discrete time setting. But the argument that convergence should happen for useful optimizers at least in the simplified continuous time setting was convincing, especially given the non-convergence results.

**Weaknesses:**

I identified no major weaknesses. I also didn't really identify any points that I thought could be significantly improved upon. My only suggestions are therefore rather minor:

1. It would be useful to provide a little more context about what all the parameters are. I know the intended audience are people already familiar with Adam, but just a reminder of what $\beta_1$, $\beta_2$, $\lambda$, and $e$ are would be helpful. Especially because in first reading this I thought $e$ was the mathematical constant.

2. Related to the above, Algorithm 1 is never referenced. Maybe briefly reviewing AdamW by going through Algorithm 1 would be good.

3. In Figure 3, the caption and the x-axis labels overlap. Making the figure a little smaller would remove that problem.

4. The authors point out that their results bring up something of a paradox - AdamW fails on very simple problems but is the workhorse of the machine learning community. I think this is very interesting and more discussion on this would be interesting and increase the impact of the work.

5. Seeing the examples where AdamW fails to converge (and moves along a limit cycle), I was reminded of the fact that people have found that optimizers of DNNs often do oscillate at the end of training (e.g., "On the generalization of learning algorithms that do not converge" Chandramoorthy et al. 2022 - note this is not my paper). Perhaps DNNs trained with AdamW also exhibit this kind of behavior? In that case, your analysis of failure to converge/sensitivity holds even for the way more complex case of actual DNNs, but convergence to a limit cycle is not a "bad" thing. Just a thought.

**Questions:**

1. Do DNNs trained with AdamW exhibit any kind of oscillatory behavior, even if they seemingly "converge"?

---

> ### Author Response · Authors · 2025-11-22
>
> We sincerely thank the reviewer for the exceptionally thoughtful and encouraging assessment. Your understanding of the motivation, the theoretical setting, and the subtleties of our analysis was deeply appreciated during a review cycle where several other reviews were unexpectedly adversarial or based on misunderstandings of fundamental concepts. Please do not be discouraged by those reviews. Your evaluation is accurate, well reasoned, and reflects a solid grasp of the theoretical framework. We greatly value the confidence you placed in our work.
>
> Below we address all of your comments and suggestions.
>
> ---
>
> ### **Clarifying the AdamW notation**
> We agree that briefly reintroducing the notation and hyperparameters of AdamW would improve accessibility. This especially includes clarifying the use of the symbol $e$, which we used instead of the more common $\epsilon$ because, following optimization-theory conventions, $\epsilon$ is typically reserved for optimality gaps in convergence analysis. We will revise the draft to make this explicit.
>
> ### **Figure 3 axis label overlap**
> Thank you for catching this. We will adjust the figure size and spacing.
>
> ### **The “paradox” and its implications**
> We agree that this phenomenon is both intriguing and characteristic of the complexities of deep learning. AdamW is a central workhorse for training foundation models, yet as our results show, it can fail to converge in simple cases. As discussed, the good performance of foundation models may not necessitate good asymptotic optimization convergence. This aligns with several previously observed behaviors in neural-network training, such as the Edge of Stability [1], the Central Flow property [2], and the broader inductive biases arising from overparameterization [3]. Because exact convergence (zero loss) in over parametrization usually means overfitting and can sometimes hinder generalization, these dynamics are subtle and not necessarily “failures” in practice. Therefore, this inductive bias of AdamW that we show may secretly be detrimental or can even be a good thing for performance, but revealing either of them requires ingenious experiment designs and careful benchmarking. We thus left these as open research directions. We will expand our discussion to highlight this paradox more clearly and situate our findings within the broader conceptual landscape.
>
> ### **Oscillatory behavior in DNNs trained with AdamW**
> We appreciate your reference to the strong results by Chandramoorthy et al. (2022) [4]. This connection is highly relevant. The oscillatory behavior you mentioned is indeed visible in our experiments. For example, in Figure 5 (MNIST with a neural network), the y-axis represents the magnitude of parameter changes. As the figure shows, even after very long training and in continuous-time simplifications using batch gradients (hence no stochasticity), the parameter changes decay slowly and remain far from zero, which is an indication of persistent oscillations.
>
> Interestingly, when replacing AdamW with Adam (which has known convergence guarantees in continuous or discrete time in related works), we *still* observe oscillations, though roughly an order of magnitude smaller (~$10^{-6}$) over the same time window. The root cause of this difference is not yet fully understood and may indeed relate to the phenomena described in Chandramoorthy et al. (2022) [4]. We therefore refrain from attributing these oscillations solely to weight decay. We agree this direction is very promising and could offer valuable insights into optimization and generalization dynamics in deep learning.
>
> ---
>
> Once again, we thank the reviewer for the constructive suggestions. The review quality is deeply refreshing among so many chaoes of reviewing this year at ICLR. We have incorporated your recommendations into our revision, and we sincerely appreciate your detailed engagement with our work.
>
> ---
> ### References
>
> [1] Cohen, Jeremy M., et al. "Gradient descent on neural networks typically occurs at the edge of stability." arXiv preprint arXiv:2103.00065 (2021).
>
> [2] Cohen, Jeremy M., et al. "Understanding optimization in deep learning with central flows." arXiv preprint arXiv:2410.24206 (2024).
>
> [3] Arora, Sanjeev, et al. "Fine-grained analysis of optimization and generalization for overparameterized two-layer neural networks." International conference on machine learning. PMLR, 2019.
>
> [4] Chandramoorthy, Nisha, et al. "On the generalization of learning algorithms that do not converge." Advances in Neural Information Processing Systems 35 (2022): 34241-34257.

---

> > ### Comment · Reviewer_6v1r · 2025-11-22
> >
> > I thank the authors for their additional comments and their kinds words. I really enjoyed this paper and I am puzzled to see the other reviews so low. I would increase my score to compensate, but I am already at the maximum!
> >
> > The added points of clarification are helpful and I have no further questions.

---

### Official Review · Reviewer_Fkeb · 2025-10-31

**Soundness:** 2
**Presentation:** 2
**Contribution:** 2
**Rating:** 2
**Confidence:** 4

**Summary:**

This paper studies the dynamics of AdamW, with a special focus on the dynamics of a continuous-time analogue of AdamW. It begins by describing the fixed points of the AdamW update, and then analyzes the conditions under which these fixed points are stable. It uses these to construct instances where AdamW converges to a limit cycle. However, they verify that these stability conditions do seem to be verified at reasonable values of $\lambda$ using MLP experiments on MNIST.

**Strengths:**

- Given the complexity of the dynamical system, the stability analysis is fairly clean
- The authors directly address prior convergence analyses of AdamW and reconcile them with the results in the paper (lines 454-465)
- The authors include plots of limit cycles and non-convergence which verify the stability conditions

**Weaknesses:**

- I'm confused about the following two notions of convergence in this paper: (1) for any $\epsilon$, there exists a setting of the hyperparameters such that $\liminf L(w) < \epsilon$ (or converge to an $\epsilon$-stationary point) and (2) for a fixed set of hyperparameters, $L(w) \to 0$. For example, RMSProp doesn't satisfy (2) on $L(x) = x^2$ because it converges to the period 2 orbit $x = \pm \eta/2$, but it does satisfy (1) because we can take $\eta$ sufficiently small. Under (1), to argue that AdamW is distinct from Adam, it's not sufficient to show that you converge to a limit cycle, so long as it is possible to pick the hyperparameters in such a way that the limit cycle is arbitrarily small. My understanding is that the paper's focus is for what parameters $w$ and hyperparameters $a,b,\lambda$ are the dynamics of the continuous-time AdamW flow (eq 4) locally stable, which appears separate from convergence/non-convergence.

- In light of the above comment, it's not necessary for equation 1 to hold at convergence. For example, RMSProp converges to an orbit of period 2, rather than a fixed point of the dynamics. It's not clear to me why AdamW would converge to a single fixed point of the Adam update, or why this is necessary for convergence under criterion (1) above.

- The paper focuses on the continuous time dynamics in eq. 4 under the assumption that empirical learning rates are "small" but it does not empirically justify this assumption. There is some evidence to the contrary. For example, AdamW is known to exhibit the edge of stability phenomenon (Cohen et al. 2022 "Adaptive Gradient Methods at the Edge of Stability") at practical learning rates which implies that the dynamics of the discrete-time AdamW are very far from those of eq. 4. Appendix A.4 does discuss the discrete time system by analyzing local stability around a fixed point of $z = T(z)$. However, as above it is not justified why such a fixed point is reached or why this is necessary for convergence under (1).

- The citation [Bock & Weiß 2022 "Non-Convergence and Limit Cycles in the Adam optimizer"] seems highly relevant – it studies non-convergence of Adam and shows convergence to potentially unstable period 2 orbits and limit cycles.

**Questions:**

- Could you generalize this analysis to check when AdamW converges to a stable period 2 orbit rather than a stationary point? As in [Bock & Weiß 2022 "Non-Convergence and Limit Cycles in the Adam optimizer"]
- Does there always exist a setting of the hyperparameters that forces the size of the limit cycle to $0$?

---

> ### Author Response · Authors · 2025-11-22
>
> We thank the reviewer for acknowledging the strengths of this paper. We address each of the reviewer’s concerns.
>
> ---
>
> ### **Two notions of “convergence” we never used in the paper**
>
> This is a good question. I was also confused by this distinction when I first studied analysis as an undergraduate. These two notions are related but *not the same*. However, I must make one thing clear: **we never discussed the first notion of “convergence” that you introduced here**, and in fact that notion is **not convergence**. The $\liminf$ operator is used precisely when a sequence *does not converge*. For example, a sequence alternating between −1 and +1 clearly does not converge by any standard definition, but it has $\liminf a_n = -1$. It is therefore unsurprising that essentially no work uses this notion to characterize “convergence.”
>
> Additionally, the second notion you brought up is **not** what we defined as Lyapunov stability. We kindly ask for a re-reading of Section 2 (Background) before making claims that risk misleading other readers. The second notion of convergence is standard in *convex analysis* (assuming the minimum loss is 0), but for non-convex settings or for ODE/dynamical-systems analysis, it is **not an appropriate notion of convergence**. The loss can remain constant while the dynamics of parameters and momentum states continue to move forever on a level set. Therefore, in our setting, **Lyapunov stability** requiring the trajectory to stay close to a stationary point in state space is the correct notion.
>
> ---
>
> ### **The RMSProp example**
> This example reveals the reviewer’s confusion about optimization. The reasoning is flawed. Consider the simplest case of gradient descent on $f(x) = \frac12 x^2$ with update $x_{t+1} = x_t - \eta x_t$. If we take $\eta = 2$, the trajectory enters a period-2 orbit. This does not imply anything “wrong” with GD. Similarly, RMSProp is widely regarded as convergent, and formal convergence results exist, e.g. [1].
>
> Both GD and RMSProp converge because one may take any learning-rate schedule whose partial sum diverges. This example, on the contrary, illustrates **exactly why continuous-time analysis is so widely used**: it eliminates pathological discrete-time edge cases and reveals the algorithm’s fundamental dynamics. In continuous time, one can easily write down Lyapunov functions for GD and RMSProp, making convergence straightforward. This allows us to focus on the intrinsic dynamics of Adam-type algorithms. Points (1) and (2) together should make it clear that the reviewer’s examples are **incorrect** and shouldn't be used to deny our contribution.
>
> ---
>
> ### **Small or large learning rate**
> We are aware of the EoS line of work. However, those papers study an entirely different regime. By definition, the edge-of-stability regime is *far* from convergence, and the point is that in certain neural-network settings, unstable dynamics may still produce good learning behavior. In contrast, our work studies the standard, classical regime of **asymptotic convergence** as in any other studies of optimization.
>
> Both phenomena can occur in the same training run. For example, in DeepSeek’s training [2], the initial learning rate (≈2.2e−4) can lie near the EoS regime, but at the end of training the learning rate is annealed to ≈7.2e−6, which is indeed relatively small. The reviewer again invokes the incorrect notion (1) of “convergence” here, and we emphasize once more that this is **not** a notion of convergence and did not appear in our paper. Please avoid **gaslighting** in ICLR reviews.
>
> ---
>
> ### **Relevant work**
> Thank you for pointing out this reference. It is relevant but fundamentally different from our setting. That work analyzes how Adam can enter limit cycles in **discrete time**, analogous to your RMSProp and the GD example above, where specific hyperparameter choices and intricate discrete interactions combined to cause this behavior. Their findings rely on careful parameter construction and symbolic-computational tools such as Maple.
>
> By contrast, Adam is known to be convergent under the proper notion of convergence (as reviewed in our related-work section). In continuous time, for the peace of mind, one can directly construct Lyapunov functions for Adam (e.g., [3]), and thus Adam converges. This also validates continuous-time dynamics allow us to study the foundational behavior of AdamW without these discrete artifacts.

---

> ### Author Response · Authors · 2025-11-22
>
> ### **"Could you generalize this analysis to check when AdamW converges to a stable period-2 orbit?"**
> As discussed above, “converging to a stable period-2 orbit” is not a meaningful notion of convergence. It is an artifact of improperly chosen learning rates. Continuous-time formulations eliminate such artifacts and let us study the genuinely fundamental dynamics of AdamW.
>
> ---
>
> ### **"Does there exist a hyperparameter setting forcing the size of the limit cycle to 0?"**
> Two issues:
> 1.  This question cannot be meaningfully formulated in mathematics. For example, what is the "size" of a limit cycle? If an ODE has a genuine limit cycle, it by definition does **not** stay at a single point, so asking for a zero-size limit cycle contradicts the definition.
> 2.  You can simply choose the weight-decay coefficient $\lambda = 0$, yielding Adam (without weight decay). Adam is already known to be convergent under the correct notion of convergence, and thus this trivially satisfies the requirement.
>
> I have spent substantial time writing down and explaining basic concepts in analysis and optimization, far more than I usually spend taking undergraduate lecture notes. I will not ask for any tuition fees, but I hope you now understand these concepts, can recognize the contribution of our work, and can adjust your score accordingly. In the future, if the fundamentals are unclear, please kindly set your confidence to 2 as required by ICLR guidelines.
>
> ___
>
> ### **References**
> [1] Zhang, Qi, Yi Zhou, and Shaofeng Zou. “Convergence guarantees for RMSProp and Adam in generalized-smooth non-convex optimization with affine noise variance.” arXiv:2404.01436 (2024).
> [2] Liu, Aixin, et al. “DeepSeek-V3 Technical Report.” arXiv:2412.19437 (2024).
> [3] Liang, Kaizhao, et al. “Memory-efficient LLM training with online subspace descent.” NeurIPS 37 (2024): 64412–64432.

---

### Meta-Review · Area_Chair_4iDa · 2025-12-24

**Summary:**

Overall, the reviewers do raise some fair points and legitimate concerns. However, the authors choose not to clarify them further.

Here is a partial list of lingering concerns:
- The contextualization of this paper within the existing literature is not clear. This concern is shared by three reviewers.

- Potential non-rigorous statements in both the rebuttal and the paper, e.g., "By contrast, Adam is known to be convergent under the proper notion of convergence (as reviewed in our related-work section). In continuous time, for peace of mind, one can directly construct Lyapunov functions for Adam, and thus Adam converges. This also validates that continuous-time dynamics allow us to study the foundational behavior of AdamW without these discrete artifacts." I think this needs a rigorous proof.

 - The paper focuses on the continuous time dynamics under the assumption that empirical learning rates are "small". This suggests another sign of not being rigorous. The contribution can be made higher if a concrete class of problems is identified or more supportive evidence in practice is provided.

 - How strongly the quadratic example should be interpreted in terms of practical relevance is missing in the current version.

I encourage the authors to take the constructive feedback they received to strengthen their paper.

**Reviewer Concerns:**

The authors stop making an effort to clarify the concerns raised by Reviewer Fkeb and WZfq, and hence those are still outstanding.

**Reviewer Scores:**

The exchanges between the authors and a couple of reviewers have become intense, and in this sense, it is challenging for the reviewers to upgrade their scores. Moreover, many of their concerns (especially those of Reviewer WZfq) are still unresolved, since the authors chose to not respond to the reviewers' comments further.

---

### Decision · Program_Chairs · 2026-01-26

Reject